# Spatial variations in snowpack chemistry, isotopic composition of $NO_3^-$ and nitrogen deposition from the ice sheet margin to the coast in West Greenland

**Chris J. Curtis[1, 5], Jan Kaiser[2], Alina Marca[2], N. John Anderson[3], Gavin Simpson[4, 5], Vivienne Jones[5], Erika Whiteford[3]**

[1]School of Geography, Archaeology & Environmental Studies, University of the Witwatersrand, Johannesburg, South Africa.
[2]Centre for Ocean and Atmospheric Sciences, School of Environmental Sciences, University of East Anglia, Norwich Research Park, Norwich, UK.
[3]Dept of Geography, Loughborough University, UK.
[4]Dept of Biology, University of Regina, Saskatchewan, Canada.
[5]ECRC, Dept of Geography, University College London, Gower Street, London, UK.

*Correspondence to*: Chris Curtis (christopher.curtis@wits.ac.za)

**Abstract.** The relative roles of anthropogenic nitrogen (N) deposition and climate change in causing ecological change in remote Arctic ecosystems, especially lakes, have been the subject of debate over the last decade. Some palaeoecological studies have cited isotopic signals ($\delta(^{15}N)$) preserved in lake sediments as evidence linking N deposition with ecological change, but a key limitation has been the lack of co-located data on both deposition input fluxes and isotopic composition of deposited nitrate ($NO_3^-$). In Arctic lakes including those in West Greenland, previous palaeolimnological studies have indicated a spatial variation in $\delta(^{15}N)$ trends in lake sediments but data are lacking for deposition chemistry, input fluxes and stable isotope composition of $NO_3^-$. In the present study, snowpack chemistry, $NO_3^-$ stable isotopes and net deposition fluxes for the largest ice-free region in Greenland were investigated to determine whether there are spatial gradients from the ice sheet margin to the coast linked to a gradient in precipitation. Late-season snowpack was sampled in March 2011 at 8 locations within 3 lake catchments in each of 3 regions (ice sheet margin in the east, the central area near Kelly Ville and the coastal zone to the west). At the coast, snowpack accumulation averaged 181 mm snow water equivalent (SWE), compared with 36 mm SWE by the ice sheet. Coastal snowpack showed significantly greater concentrations of marine salts ($Na^+$, $Cl^-$, other major cations), ammonium ($NH_4^+$; regional means 1.4-2.7 µmol $L^{-1}$), total and non-sea salt sulfate ($SO_4^{2-}$; total 1.8-7.7, non-sea salt 1.0-1.8 µmol $L^{-1}$) than the two inland regions. Nitrate (1.5-2.4 µmol $L^{-1}$) showed significantly lower concentrations at the coast. Despite lower concentrations, higher precipitation at the coast results in greater net deposition for $NO_3^-$ as well as $NH_4^+$ and non-sea salt sulfate (nss-$SO_4^{2-}$) relative to the inland regions (lowest at Kelly Ville 6, 4 and 3; highest at coast 9, 17 and 11 mol $ha^{-1}$ $a^{-1}$ of $NO_3^-$, $NH_4^+$ and nss-$SO_4^{2-}$ respectively). The $\delta(^{15}N)$ of snowpack $NO_3^-$ shows a significant decrease from inland regions (-5.7 ‰ at Kelly Ville) to the coast (-11.3 ‰). We attribute the spatial patterns of $\delta(^{15}N)$ in West Greenland to post-depositional processing rather than differing sources because of 1) spatial relationships

with precipitation and sublimation, 2) within-catchment isotopic differences between terrestrial snowpack and lake-ice snowpack, and 3) similarities between fresh snow (rather than accumulated snowpack) at Kelly Ville and the coast. Hence the $\delta(^{15}N)$ of coastal snowpack is most representative of snowfall in West Greenland, but after deposition the effects of photolysis, volatilization and sublimation lead to enrichment of the remaining snowpack with the greatest effect in inland areas of low precipitation and high sublimation losses.

## 1. Introduction

In recent years it has been demonstrated that anthropogenic nitrogen deposition, primarily from fossil fuel combustion, has reached areas very remote from the original sources, including high latitude sites in the Arctic. Evidence includes contemporary deposition monitoring (AMAP, 2006), the snowpack record of the Greenland ice sheet (Hastings et al., 2009) and palaeolimnological records in Arctic lakes (Holtgrieve et al., 2011). However, contemporary deposition data are sparse in such remote areas due to logistical and cost limitations. According to AMAP (2006), "more observations for $NO_3^-$ in air and precipitation are required to better understand the development of $NO_3^-$ pollution in the Arctic" and Greenland is a striking example of the paucity of data. In addition, stable isotopes of $NO_3^-$ have been used to understand both temporal changes and spatial patterns in N deposition as well as links to ecological changes recorded in Arctic lake sediments. Holtgrieve et al. (2011) asserted that $\delta(^{15}N)$ records in lake sediments indicate a "coherent signal of anthropogenic nitrogen deposition to remote watersheds", yet a central problem remains why some remote lakes record a 20[th] century $^{15}N$ depletion signal, attributed to atmospheric inputs of anthropogenically produced $NO_3^-$, whilst others (often neighbouring sites) do not. This is an important issue as $\delta(^{15}N)$ in lake sediments offers one of the few means of identifying a nitrogen effect on remote lakes over historic timescales and is central to the debate over the relative roles of of climate change and nitrogen deposition in driving ecological change in remote lakes (Catalan et al. 2009; Wolfe et al. 2013). Stable isotope data are even more restricted than precipitation chemistry in the Arctic despite their value for understanding pollutant pathways and ecological impacts, with most published data derived from studies in the centre of the Greenland ice sheet (Hastings et al., 2009; Fibiger et al., 2016) or from Svalbard (Heaton et al., 2004; Tye and Heaton, 2007; Björkman et al., 2014).

The largest ice-free region of Greenland is found in the south-west, where a great number of lakes have been the subject of several limnological and palaeolimnological studies. The region between the edge of the ice sheet, the key international airport hub at Kangerlussuaq and the coastal town of Sisimiut, was selected for an integrated study into the potential effects of nitrogen deposition on Arctic lakes (Figure 1) without the confounding effects of climate warming, since there was no significant warming trend in the region for most of the 20[th] century (Hanna et al., 2012). This region contains lakes showing very different $\delta(^{15}N)$ sediment records between coastal sites and inland regions (e.g. Perren et al., 2009; Reuss et al., 2013), which we hypothesize may be driven by differences in the net delivery of $NO_3^-$ in deposition. This study presents a first

attempt to characterise the chemistry and isotopic composition of $NO_3^-$ inputs across an assumed deposition gradient from the ice sheet margin to the coast.

Around half of precipitation in West Greenland falls as snow (e.g. 45 % at Sisimiut and 52 % at Kangerlussuaq from 1994-1997; Yang et al., 1999, and 40% at the ice sheet in 2011-13; Johansson et al., 2015). Hence snowpack chemistry (if unchanged following deposition) can provide the data required for estimating annual deposition of pollutants in remote Arctic regions where regular deposition monitoring is not possible due to logistical and financial constraints. In high snowfall regions with a fairly continuously accumulating snowpack, late season snowpack may provide a good estimate of total deposition inputs over the snow season, which may cover more than 6 months in high altitude or high latitude sites (e.g. Rockies – Turk et al., 2001; Ingersoll et al., 2008; Williams et al., 2009). However, in West Greenland the inland areas experience very low precipitation inputs and sublimation of accumulated snowpack is also important (Johansson et al., 2015). Annual mean precipitation at Sisimiut from 2001-2012 was 631 mm while at Kangerlussuaq it was 258 mm (Mernild et al., 2015). Much greater accumulation of snowpack also occurs in the coastal areas, so it is expected that there is a gradient of precipitation, snowpack accumulation and resultant deposition of pollutants from the interior ice sheet margin to the coast.

**1.1 Precipitation chemistry, nitrogen deposition and post-depositional processing in the Arctic**

Most Arctic precipitation chemistry and acid deposition fluxes ($NO_3^-$, $NH_4^+$ and $SO_4^{2-}$) have been reported under the auspices of the Arctic Monitoring and Assessment Programme (AMAP, 2006). Spatial assessments of both concentrations (Hole et al., 2006a) and sources (Hole et al., 2006b) indicate spatial heterogeneity across the Arctic, but mostly with very low concentrations and deposition fluxes across much of the region including Greenland, relative to more industrialised regions further south. Sources of acid deposition precursors ($NO_x$, $SO_2$) within the Arctic are restricted to a few major point sources, such as large smelters on the Kola Peninsula and Norilsk in the Russian Arctic, with other poorly quantified emissions associated with shipping and oil or gas exploration (Hole et al., 2006b). Pollution is also transported into the Arctic from all northern hemisphere continents, but predominantly from Europe and North America (Hole et al., 2006b). With respect to anthropogenic nitrogen deposition and isotopic composition of $NO_3^-$ in the Arctic, the majority of published studies relate to seasonal snowpack and ice core records on the Greenland ice sheet, where the fate of deposited $NO_3^-$ is determined by a complex range of processes.

The processing of $NO_3^-$ in deposited snowpack, termed post-depositional processing, occurs at the air-snow interface and may entail losses and in situ cycling of $NO_3^-$, with different impacts on both net deposition fluxes and isotopic fractionation depending on their relative importance (Frey et al., 2009; Geng et al., 2015; Fibiger et al., 2016). Nitrate may be released back to the atmosphere by desorption and evaporation as $HNO_3$, often termed 'physical' losses (Mulvaney et al., 1998; Berhanu et al., 2015), or by photolysis (sometimes referred to as photodenitrification) (Frey et al., 2009). Photolysis of

snowpack $NO_3^-$ by UV radiation produces $NO_x$, which may then undergo various processes, which differ in relative importance depending on local conditions. $NO_x$ may be

1. re-emitted from the snowpack and transported away from the area, depending on wind speed;

2. redeposited by dry deposition;

3. reoxidised back to $NO_3^-$ and redeposited (re-adsorption or dissolution) (Frey et al., 2009).

Erbland et al. (2015) define "$NO_3^-$ recycling" as the net effect of $NO_3^-$ photolysis (producing $NO_x$), following atmospheric processing and oxidation to form atmospheric $NO_3^-$, and the local redeposition (wet or dry) and export of products. Recycling may also include redeposition of directly emitted $HNO_3$ (Erbland et al., 2013). Hence both physical and photolytic processes may lead to effective net losses of $NO_3^-$ from the snowpack if products are transported away from the area, but a

proportion may be recycled and hence does not result in net removal from the snowpack, although such recycling can progressively modify isotopic signatures of the $NO_3^-$.

Photolysis is associated with large fractionation of both N ($^{15}\varepsilon$ between -48 and -56 ‰) and O ($^{18}\varepsilon$ = -34 ‰) which both tend to increase $\delta(^{15}N)$ and $\delta(^{18}O)$ in the remaining snowpack $NO_3^-$ if the $NO_x$ produced is removed from the system (Frey et al., 2009; Erbland et al., 2013; Berhanu et al., 2015; Geng et al, 2015). In situ recycling of $NO_3^-$ can also reduce $\delta(^{18}O)$ and

$\Delta(^{17}O)$ due to oxygen isotope exchange with water (Frey et al., 2009; Shi et al., 2015), which has a different isotopic signature than atmospheric oxidants. This means that the negative $^{18}\varepsilon$ is not expressed in the residual snow $NO_3^-$ and, in fact, the apparent overall oxygen isotope fractionation can be positive (between 9 and 13 ‰, Berhanu et al., 2015). However, the depth-integrated $\delta(^{15}N)$ remains constant if there is no net loss of $NO_3^-$, hence $\delta(^{15}N)$ is deemed a more reliable indicator of net post-depositional losses than oxygen isotopes (Geng et al., 2015; Zatko et al., 2016). Much smaller (only slightly

negative) fractionation constants for other processes have been derived, e.g. physical release of $NO_3^-$ (evaporation) but studies in the Antarctic by Erbland et al. (2013) found different experimental values at different temperatures and hence these factors are not generally transferable to regions with differing climatic regimes.

Antarctic studies have generally found photolysis to be the dominant driver of $NO_3^-$ remobilisation and isotopic fractionation,

while acknowledging that physical processes could play a greater role in coastal and other regions (Erbland et al., 2013; Berhanu et al., 2015). Erbland et al (2013, 2015) working in Antarctica found that changes in $\delta(^{18}O)$ and $\Delta(^{17}O)$ through $NO_3^-$ loss and recycling were much less pronounced than $\delta(^{15}N)$ and either slightly positive or not significantly different from zero. Similar results for $\Delta(^{17}O)$ were also found experimentally by McCabe et al. (2005) and Berhanu et al. (2015). Erbland et al (2013) suggested that the small apparent oxygen isotope fractionation factors in their coastal Antarctic snowpack could

indicate a greater role for physical $NO_3^-$ release, which does not entail oxygen exchange. Zatko et al. (2016) demonstrated that recycling of snow $NO_3^-$ in Greenland, where $NO_3^-$ spends a much shorter time in the photic zone, is much less than in Antarctica. They assumed that wet deposited $NO_3^-$ is more likely to be embedded in the interior of snow grains whereas dry deposited $NO_3^-$ on the grain surface should be more photolabile, so that in situ recycling is also a function of the form (wet vs. dry) of $NO_3^-$ deposited.

Here we describe the spatial variation in total inorganic nitrogen (TIN: $NO_3^- + NH_4^+$) and sulfur deposition in snowpack and the isotopic signature of snowpack $NO_3^-$ ($\delta(^{15}N)$, $\delta(^{18}O)$ and $\Delta(^{17}O)$) for three regions in West Greenland. We use the strong precipitation gradient to test the hypothesis that the delivery of TIN deposition will differ from the ice sheet margin to the coastal region while simultaneously determining differences in the isotopic composition of $NO_3^-$ due to post-depositional processing.

## 2. Methods

### 2.1 Site selection

As part of a wider study of the ecology and palaeolimnology of low Arctic lakes, deposition study sites were based in three clusters of lake catchments along an assumed deposition gradient from the ice sheet margin to the coast (Figure 1, Table 1), hereafter referred to as ice sheet, Kelly Ville and coastal sites. Three lake catchments were chosen within each region on the basis of previous studies and suitability for (palaeo-)limnological studies reported elsewhere. Five replicated late season snowpack samples were collected within the terrestrial part of each catchment, with a further three replicates obtained from the snowpack on the frozen lake surface. Hence for the purposes of the present study considering spatial patterns, eight samples each from three lake catchments are considered to represent 24 replicated samples within each region. All catchments are located within a narrow latitudinal band around 67° N with a maximum difference in latitude of only 0.2° (22 km). The maximum distance between sites is 153 km (152 km in east-west direction). The distances between regions are much greater than the distances between lake catchments within each region. The central Kelly Ville sites are at least 98 km from the closest coastal site and 33 km from the closest ice sheet site. Within each region, the largest distance between sites is 12 km at the coast, 8 km at Kelly Ville and 5 km at the ice sheet.

### 2.2 Snowpack estimation and sampling

Snowpack depth and density were measured during repeat traverses of each catchment along a grid-based pattern to obtain a spatial coverage of 50-150 measurements per catchment. Depth was measured every 100 m with a graduated pole while density was estimated by taking a snow core of known volume using a 37 mm internal diameter plastic pipe at every 5th measurement point and weighing in the field using a spring balance. Snowpack sampling locations were selected to obtain representative spatial coverage within each lake catchment, recognising the spatial variations in aspect, altitude and snowpack depth where snowpack coverage was unevenly distributed within catchments. Within each catchment, samples were obtained from upper, mid-level and lakeside elevations and different aspects, but logistical constraints limited sampling

to just five locations. In addition, three lake snowpack samples on top of the lake ice were obtained from equally spaced locations along the longest axis of the frozen lake. Hence eight samples per catchment were collected, but comparisons of terrestrial snowpack and snow accumulated on lake ice were also possible.

Snowpack was sampled according to USGS ultra-clean protocols (Clow et al., 2002; Ingersoll et al., 2008, 2009). In summary, all sampling equipment and sample bags were triple rinsed with distilled deionised water (DDIW), with field blanks obtained by rinsing off the sampling shovel and scoop into a clean sample bag with a DDIW wash bottle in the field. Depth-integrated snow samples were collected with a polycarbonate scoop and kept frozen in clean polyethylene bags until processed in the laboratory. The whole snowpack was sampled down to ground level and hence represents an integrated

sample incorporating the net effects of post-depositional processing over the winter season. Fresh latex gloves for each sample were worn at all times while sampling and processing in the laboratory. Back in the laboratory, snow samples were allowed to thaw at room temperature overnight and then filtered through 0.45 µm nylon membrane filters (Millipore) prior to storage and freezing in 125 ml ultra-clean LDPE bottles. Samples were kept frozen and transported back to the isotope laboratory at UEA where they were stored frozen prior to analysis.

Sampling was carried out in the late winter period to capture as much of the accumulated snowpack as possible without the risk of substantial snowmelt occurring (cf. de Caritat et al., 2005); all total snowpack samples were collected between 22[nd] March and 1[st] April 2011. Snowpack profile temperature and physical description were noted as per Ingersoll et al. (2009) for assessment of snowpack status and whether melt was in progress. In addition, to assist with bulk deposition estimates, ad

hoc sampling of rainfall and fresh falling snow was carried out on numerous occasions during field campaigns in each region during 2011 and 2012.

### 2.3 Chemical and isotopic analysis

The nitrogen ($^{15}$N/$^{14}$N) and oxygen ($^{18}$O/$^{16}$O, $^{17}$O/$^{16}$O) isotope ratios of $NO_3^-$ were determined using the denitrifier method

(Casciotti et al., 2002; Kaiser et al., 2007). The isotope ratios are expressed as relative isotope ratio differences (isotope deltas) with respect to the international reference materials Air-$N_2$ for nitrogen isotopes and Vienna Standard Mean Ocean Water (VSMOW) for oxygen isotopes, e.g.

$$\delta(^{15}N/^{14}N, NO_3^-) = \frac{R_{sample}(^{15}N/^{14}N, NO_3^-)}{R_{reference}(^{15}N/^{14}N, NO_3^-)} - 1 \quad (1)$$

Instead of the complete quantity symbols (i.e. $\delta(^{15}N/^{14}N, NO_3^-)$ etc.) we use the short-hand notation $\delta(^{15}N)$, $\delta(^{17}O)$ and

$\delta(^{18}O)$. Since $\delta(^{17}O)$ and $\delta(^{18}O)$ are highly correlated, we use the $^{17}O$ excess, $\Delta(^{17}O)$, instead of $\delta(^{17}O)$ and define it (following Kaiser et al., 2007) as:

$$\Delta(^{17}O) = \frac{1 + \delta(^{17}O)}{[1 + \delta(^{18}O)]^{0.5279}} - 1 \quad (2)$$

$\delta(^{15}N)$ and $\delta(^{18}O)$ values were determined using the standard denitrifier method with $N_2O$ as analyte gas (Casciotti et al. 2002). $\Delta(^{17}O)$ was determined using the thermal decomposition method with $O_2$ as analyte gas (Kaiser et al. 2007). $\delta(^{15}N)$
values have been corrected for isobaric interference of $N_2^{17}O$ (Kaiser & Röckmann, 2008), using the $\Delta(^{17}O)$ measurements. The international reference material IAEA-NO-3 was used for calibration of the delta values, using $\delta(^{15}N) = 4.7$ ‰ (vs. Air-$N_2$), $\delta(^{18}O) = 25.61$ ‰ (vs. VSMOW) and $\delta(^{17}O) = 13.18$ ‰ (vs. VSMOW) (Kaiser et al. 2007), giving $\Delta(^{17}O) = -0.25$ ‰. In addition, the reference materials USGS 34 ($\delta(^{18}O) = -27.93$ ‰, $\Delta(^{17}O) = 0.04$ ‰) and USGS 35 ($\delta(^{18}O) = 57.50$ ‰, $\Delta(^{17}O) = 20.88$ ‰) were used to correct the measurements for oxygen isotope scale contraction. The analytical precision
(repeatability) based on repeat sample analysis was 0.2 ‰ for $\delta(^{15}N)$, 0.5 ‰ for $\delta(^{18}O)$ and 0.3 ‰ for $\Delta(^{17}O)$ (10 nmol $NO_3^-$). The standard deviations represent analyses in duplicate.

### 2.3.1 Base cations, $SO_4^{2-}$, $Cl^-$

Chloride ($Cl^-$) and $SO_4^{2-}$ concentrations were measured using ion chromatography on a Dionex ICS-2000 system (Thermo
Fisher Scientific) comprising a Dionex Ion Pac AG18 guard column (50 mm x 2 mm), a Dionex Ion Pac AS18 analytical column (250 mm x 2 mm), isocratic elution with potassium hydroxide (KOH) at 24 mmol $L^{-1}$, flow rate: 0.250 ml $min^{-1}$, column temperature: 30 °C, with suppressed conductivity detection. Standards and ultrapure water blanks (18 MΩ cm, Purelab Ultra) were analysed at the beginning, in the middle and at the end of each sample batch for the calibration of the instrument and to account for any instrument drift during the run. The data were processed using Chromeleon software 6.8
(Thermo Fisher Scientific). The relative analytical precision (repeatability) based on repeat sample analysis was 2 % for both $Cl^-$ and $SO_4^{2-}$. Major cations, $Na^+$, $K^+$, $Mg^{2+}$ and $Ca^{2+}$ were determined using inductively coupled optical emission spectroscopy (ICP-OES; Varian Vista-Pro). Mean instrumental detection limits were 0.1, 0.01, 0.01 and 0.1 µM, respectively, with a relative instrumental precision of 10 %.

**2.3.2 Nutrients ($NO_2^-$, $NO_3^-$, $NH_4^+$, phosphate): autoanalyser**

Nitrite ($NO_2^-$), $NO_3^-$ (after cadmium reduction to $NO_2^-$), $NH_4^+$ and phosphate ($PO_4^{3-}$, $HPO_4^{2-}$, $H_2PO_4^-$) concentrations were determined colorimetrically using a Skalar San$^{++}$ autoanalyser. Detection limits of 0.1 µM (for $NO_3^-$ and $NH_4^+$) and 0.01 µM (for $NO_2^-$ and phosphate) were achieved, with a relative instrumental precision of 4 %.

## 2.4 Statistical analysis

Generalized linear mixed models (GLMM) were used to investigate how snowpack chemistry and isotope variables varied between regions (15 catchment replicates plus 9 lake snow replicates; n=24 per region). A random intercept was included in the model to account for clustering in the data at the catchment level. Snowpack chemistry variables were modelled using a Gamma GLMM with log link function to account for non-constant variance. Stable isotope variables were modelled using a Gaussian GLMM with identity link function. Additional models explored differences in snowpack chemistry and stable isotopes between samples of catchment snow and snow over lake ice by including sample type, region, and their interaction terms in the fitted models.

Post-hoc pairwise comparison of the GLMM-estimated regional means was performed using Tukey contrasts and the generalized linear hypothesis testing (GLHT) framework. For models of differences in snow sample type (terrestrial versus lake ice) post-hoc comparisons were restricted to comparison of sample type within region using appropriate contrast matrices.

All statistical analyses were performed using the R statistical language (version 3.3.2; R Core Team, 2016) with the lme4 package (version 1.1.12; Bates et al, 2015) for fitting GLMMs and the multcomp package (version 2.4.6, Hothorn et al, 2008) for GLHT post-hoc comparisons.

## 3. Results

### 3.1 Catchment scale snowpack depth and SWE estimation

Estimates of snowpack depth and snow water equivalents (SWE) are presented in Table 2. Note that snow water equivalent calculations were carried out for the subset of points at which snow mass was measured in the snow tube and then corrected for mean snow depth across each catchment based on the much larger number of snow depth measurements. Snow density ranged from 0.20 to 0.34 g cm$^{-3}$, remarkably similar to the datasets used in the wide-ranging snow depth-SWE study of Sturm et al. (2010) (0.21-0.34 g cm$^{-3}$).

Snow depth measurements confirm that there is a major difference in snowpack accumulation from 16 to 20 cm mean catchment snow depth (max. 103 cm; overall 35.7 mm SWE) close to the ice sheet, up to 50-74 cm mean catchment snow depth (max. 260 cm; overall 180.8 mm SWE) at the coast. Depth and SWE are slightly higher at the central Kelly Ville catchments relative to the ice sheet sites, but all are still much lower than the coastal sites. At the two inland regions, snow cover was much more patchy than at the coast, where continuous cover was found except on the steepest slopes (Figure 2).

## 3.2 Snowpack sampling and chemistry

Details of snowpack samples for chemical and isotopic analysis are provided in Table 3a (terrestrial) and 3b (lake ice snowpack). At the ice sheet, sampled terrestrial snowpack varied from 24 to 103 cm depth while sampled lake ice snowpack reached only 23 cm maximum depth, reflecting the heterogeneous snow distribution around the catchment and suggesting wind redistribution of the snow. Sampled terrestrial snowpack depth had a smaller range at the Kelly Ville catchments, from 23 to 65 cm, and again the lake ice snow depth was much shallower, reaching only 30 cm maximum depth. At the coastal sites, terrestrial snowpack depths from 28 to 240 cm were sampled while lake ice snow ranged from 19 to 60 cm depth. During the sampling period, air temperatures were all well below 0 °C at the ice sheet and Kelly Ville sites, reaching a maximum of -6 °C. However, during sampling of the coastal snowpack, air temperatures were slightly above freezing, suggesting that some snowmelt may have been occurring during the sampling period (Table 3a-b). Snowpack temperatures were only just below zero at the coastal sites and melting was observed in the lake ice snowpack at site AT1.

## 3.3 Regional comparison of aggregated snowpack data

Concentrations of major ions in West Greenland snowpack are very low (<5 µmol L$^{-1}$) except for $Mg^{2+}$, $SO_4^{2-}$ and the sea-salt associated ions $Na^+$ and $Cl^-$ (Table 4). Analysis of the aggregated snowpack data shows that there are significant differences for most measured analytes except $NO_2^-$ between the coastal sites and both the Kelly Ville and ice sheet margin sites, while snowpack composition is very similar between the two inland regions (Table 4). For $NH_4^+$, $SO_4^{2-}$, $Cl^-$ and all base cations, concentrations are significantly higher in coastal snowpack than in the Kelly Ville or ice sheet regions (except for no significant differences between the coast and ice sheet sites for $NH_4^+$ and $Ca^{2+}$). The sea-salt-associated ions $Na^+$ and $Cl^-$ are highly correlated (r=0.999, p<0.01) and concentrations are an order of magnitude greater at the coast than inland. $Mg^{2+}$, $K^+$ and $SO_4^{2-}$ are also very highly correlated with $Cl^-$ (r>0.95, p<0.01). The nutrients $NO_3^-$ and $PO_4^{3-}$ show an opposing pattern, with significantly lower concentrations in coastal snowpack than at inland sites and weak, negative correlations with $Cl^-$ ($NO_3^-$: r=-0.392, p<0.01; $PO_4^{3-}$: r=-0.277, p<0.05).

In order to investigate the influence of non-sea salt atmospheric sources of ions, the proportion of sea salt contributions was subtracted using $Cl^-$ as a tracer of sea salt inputs and the relatively constant ionic proportions of major ions in seawater (Henriksen & Posch, 2001). For non-sea salt (nss) $SO_4^{2-}$, the pattern of snowpack concentrations is similar to total measured values (Table 4), with the highest mean concentrations at the coast which are significantly greater than inland, although mean non-sea salt concentrations are all very low (1.0-1.8 µmol L$^{-1}$). Non-sea salt $Mg^{2+}$ is significantly lower at Kelly Ville than elsewhere (Table 4), while non-sea salt $Ca^{2+}$ increases significantly from the coast towards the ice sheet, presumably

due to wind-blown minerogenic sources. There are no significant differences for non-sea salt $K^+$ and negative values for non-sea salt $Na^+$ suggest either non-sea salt sources of $Cl^-$ or snowpack losses of $Na^+$, perhaps through preferential elution pathways.

For nutrients, concentrations in seawater are assumed to be negligible hence snowpack concentrations are assumed to be due entirely to non-sea salt atmospheric inputs. Nitrate concentrations are very low at all sites but significantly lower (p<0.0001) at the coast (mean 1.5 µmol $L^{-1}$) than at Kelly Ville (mean 2.3 µmol $L^{-1}$) or the ice sheet (mean 2.4 µmol $L^{-1}$), with no significant difference between the inland regions. Mean $NH_4^+$ concentrations are also low, but significantly higher in coastal snowpack (2.7 µmol $L^{-1}$) than Kelly Ville (1.4 µmol $L^{-1}$). Nitrite levels were negligible in all regions.

## 3.4 Catchment snowpack versus lake ice snowpack

Exploratory data analysis of separate terrestrial snowpack (n=5 per catchment) and lake ice snowpack (n=3 per catchment) was carried out to determine whether there were within-catchment differences in snowpack chemistry between catchment slopes (with very heterogeneous snow cover) and the relatively homogenous snow cover on the frozen lake. Mean lake ice

snowpack concentrations were higher than terrestrial snowpack concentrations for all ions except for a few cases with very low ionic concentrations of <3 µmol $L^{-1}$, with the difference being most pronounced at the coastal sites (Table SI 1). While very few of these differences were statistically significant except at the coast (p<0.05 for all plotted ions), the pattern of higher lake ice snowpack concentrations was remarkably consistent (Figure 4). For $NO_3^-$, significantly higher concentrations were found in lake ice snowpack at the coast (p=0.028) and Kelly Ville (p=0.0042) but the difference was not quite

significant for the ice sheet sites (p=0.0731) – noting that only 3 replicated lake ice snowpack samples resulted in large standard errors.

The only sites where melting snow was observed during sampling were at the coast, raising the possibility that in coastal catchments, the snowpack on lake ice could be the recipient of meltwater drainage from catchment slopes whereby

preferential elution from catchment snowpack could explain the higher concentrations in the lake snow. Alternatively, losses of some ions such as $NO_3^-$ back to the atmosphere may be greater on catchment slopes (see Discussion below considering stable isotope data). Despite differences between catchment snow and lake ice snowpack, the general patterns of higher major ion concentrations at the coast but lower nutrients is repeated (as seen in the aggregated catchment data).

### 3.5 Stable isotopes

For aggregated snowpack $\delta(^{15}N)$ there are significant differences ($p<0.001$) between coastal and inland regions (as with major ion chemistry) with the lowest mean values at the coast (-11.3 ‰) and the highest at Kelly Ville (-5.7 ‰) (Table 4). There are no significant differences in $\delta(^{18}O)$, with mean values of 81.7-83.3 ‰ across all regions. None of the measured isotopes show significant differences between inland regions. Values of $\Delta(^{17}O)$ in catchment snowpack range from 30.8 ‰ at the coast to 34.4 ‰ at Kelly Ville, again with significantly lower values in coastal snowpack than for inland regions ($p<0.01$). Within regions, no significant differences in $\delta(^{15}N)$, $\delta(^{18}O)$ or $\Delta(^{17}O)$ are observed between terrestrial and lake ice snowpack, although there is a consistent pattern within each region of generally lower $\delta(^{15}N)$ in lake ice snowpack (Fig. S.I. 1).

### 3.6 Deposition estimates

In addition to the analysis of accumulated snowpack in the study regions, ad-hoc sampling of fresh snow and rainfall was carried out on numerous occasions during late winter, summer and autumn field campaigns. Unfortunately, most of the bulk deposition samples collected were subject to major contamination by bird strikes by the northern wheatear (*Oenanthe oenanthe*) which finds any prominent vertical structures in the low Arctic scrub an irresistible vantage point, despite attempts to fit various configurations of bird deterrent devices. However, a small number of uncontaminated rainfall samples were collected along with fresh snowpack samples where a surface accumulation of falling snow was collected within a few hours of being deposited (Table 5).

Fresh snow collected from coastal sites in 2011 had slightly higher nutrient concentrations compared to catchment snowpack but had very low concentrations of sea salt related ions, indicating that in the fresh falling snow the influence of seasalt inputs was minimal. Presumably marine aerosols accumulate in the snowpack over winter, which may explain the higher concentrations of $NH_4^+$ as well as $SO_4^{2-}$ at the coast. Fresh snow collected at Kelly Ville in April 2011 also had slightly higher concentrations of $NO_3^-$ and $NH_4^+$ than regional snowpack but had very similar major ion concentrations. Four rainfall samples from the Kelly Ville region in 2011 had variable concentrations of $NO_3^-$ (1.3- 7.8 µmol L$^{-1}$) but the mean of 4.0 µmol L$^{-1}$ was higher than the regional snowpack (mean 2.3 µmol L$^{-1}$) while $NH_4^+$ concentrations in the rainfall were slightly lower than the snowpack (Table 5). Several rainfall samples were also collected from the ice sheet region, again showing ca. 50% higher mean $NH_4^+$ (3.1 µmol L$^{-1}$) and $NO_3^-$ (3.4 µmol L$^{-1}$) than the snowpack.

Logistical challenges prevented the routine monitoring of non-snowpack precipitation, and while around half of annual precipitation falls as snow in West Greenland, this does mean that annual deposition fluxes can only be estimated using best available data. In this region, we assume that snowpack concentrations of atmospherically derived ions are representative of

total annual precipitation and hence can obtain a first approximation of deposition fluxes by using mean snowpack solute concentrations with measured annual precipitation data at Sisimiut and Kangerlussuaq (Mernild et al., 2015) and scaled for ice sheet data at SS903 from 2011-12 (Johansson et al., 2015). Estimated deposition loads based on mean snowpack chemistry and mean 2001-2012 precipitation levels for Sisimiut (coast) and Kangerlussuaq (inland regions) are shown in

Table 6.

While $NO_3^-$ concentrations are lower at coastal sites than inland, higher precipitation levels at the coast lead to 18-62 % greater $NO_3^-$ deposition than inland. For $NH_4^+$ where coastal sites have both higher concentrations and higher precipitation, estimated deposition loads are around 3-5 times higher at the coast than inland. Overall, total inorganic N deposition at the

coast is therefore estimated to be 1.9-2.8 times higher than for the inland regions. For non-sea salt $SO_4^{2-}$ the deposition at the coast is also 2.7-4.4 times higher than inland.

## 4. Discussion

## 4.1 Precipitation chemistry

The gradient in precipitation from the coast to the ice sheet has been attributed by Mernild et al. (2015) to katabatic winds moving downslope from the ice sheet interior, distance from oceanic moisture sources and orographic enhancement by coastal mountains, all contributing to much greater precipitation at the coast relative to areas further inland towards the ice sheet. There is a major difference in the chemistry of snowpack from inland to the coast which is primarily driven by the greater influence of marine inputs (sea spray and aerosols) at the coast, clearly shown by highly elevated concentrations of

$Na^+$ and $Cl^-$ (cf. coastal snowpack in Svalbard studied by Tye & Heaton, 2007) but also by separate gradients in atmospheric pollutant deposition. Concentrations of $NH_4^+$ ($p=0.0017$ for coast-Kelly Ville) and nss-$SO_4^{2-}$ ($p<0.05$) are greater in coastal snowpack than inland, but concentrations of $NO_3^-$ ($p<0.0001$) are lower at the coast. Hence there is clearly an interaction between dilution effects of greater precipitation at the coast and differential pollutant inputs and presumably pathways from inland to coastal regions (see below).

Snowpack solute concentrations in this study are comparable to values recorded in studies on the Greenland ice sheet. Fischer et al. (1998a) studied chemistry of recent firn along ice sheet transects and recorded a range of 110-150 ng g$^{-1}$ (1.8-2.4 µmol L$^{-1}$) $NO_3^-$ and 70-110 ng g$^{-1}$ (0.7-1.1 µmol L$^{-1}$) $SO_4^{2-}$ for central Greenland. Burkhart et al. (2004) recorded a mean $NO_3^-$ of 2.9 µmol L$^{-1}$ in surface snow at Summit from 1997-1998 (range 0.4-34.4 µmol L$^{-1}$) while a later study indicated

recent peaks of 2-5 µmol L$^{-1}$ in 6 ice cores (Burkhart et al., 2006). Dibb et al. (2007) studied daily snowpack chemistry at Summit from 1997-98 and then from August 2000-August 2002, and recorded overall mean concentrations of 0.5 µmol L$^{-1}$

for $NH_4^+$ (monthly mean range 0.1-1.4 µmol $L^{-1}$) and 3.2 µmol $L^{-1}$ for $NO_3^-$ (monthly mean range 1.3-6.7 µmol $L^{-1}$). Mean $SO_4^{2-}$ was 0.7 µmol $L^{-1}$ (monthly mean range 0.2-2.3 µmol $L^{-1}$) while mean $Na^+$ was 0.4 µmol $L^{-1}$ and $Cl^-$ was 0.8 µmol $L^{-1}$. In equivalence terms, $NO_3^-$ constituted the dominant ion in fresh snow at Summit while sea salt ion concentrations were negligible, much lower than the terrestrial snowpack in our study and reflecting the much greater distance from the coast of

the ice sheet studies. Dibb et al. (2007) found that $NO_3^-$ was the only ion having a higher concentration in fresh snow compared with buried layers, but the difference, presumably due to post-depositional processing, was only 9 %. More recent samples at Summit showed mean concentrations of 2.8 and 5.2 µmol $L^{-1}$ for the 2010 and 2011 seasons (Fibiger et al., 2016). Our results are also within the range of other studies of Arctic precipitation and ice cores. Kekonen et al. (2002) recorded peak concentrations of 3-4 µmol $L^{-1}$ for $NO_3^-$ and 4-5 µmol $L^{-1}$ for $NH_4^+$ during the 1980s in Svalbard ice cores. Tye &

Heaton (2007) found concentrations of 1.7-3.1 µmol $L^{-1}$ $NO_3^-$ and 1.2-1.7 µmol $L^{-1}$ for $NH_4^+$ in Svalbard snowpack. In the AMAP synthesis of Arctic precipitation chemistry data (Hole et al., 2006a), $NO_3^-$ concentrations for the period 1980-2005 ranged from 0-10 µmol $L^{-1}$ but the great majority of annual mean values were <4 µmol $L^{-1}$. However, the majority of stations showed higher winter than summer precipitation concentrations, unlike our study where analysis of ad hoc rainfall samples suggested higher concentrations of $NO_3^-$ and $NH_4^+$ in rainfall relative to snowpack (Table 5). Sulfate concentrations were

much more spatially variable, but the great majority of annual mean concentrations were <10 µmol $L^{-1}$ and some regions showed higher summer than winter concentrations (Hole et al., 2006a). De Caritat et al. (2005) carried out a wide-ranging snapshot survey of Arctic snowpack chemistry and found snowpack concentrations at Pittufik (NW Greenland) of 5.7-8.9 µmol $L^{-1}$ for $NO_3^-$ and 9.9-12.8 µmol $L^{-1}$ for $SO_4^{2-}$, with median values across their Arctic survey of 3.5 and 9.9 µmol $L^{-1}$ respectively. Jaffe and Zukowski (1993) recorded 2.6 and 1.9 µmol $L^{-1}$ for $NO_3^-$ and $SO_4^{2-}$ in Alaskan snowpack.

Hence the chemistry of West Greenland snowpack is comparable to the Greenland ice sheet and other areas of the Arctic remote from pollution sources, but with lower acid anion concentrations than more polluted regions of the Arctic such as parts of the Russian Federation and NW Europe (De Caritat et al., 2005; Hole et al., 2006b). Snowpack concentrations in this part of the Arctic are also generally lower than those recorded in remote alpine systems such as the Rockies (e.g. 10-12 µmol

$L^{-1}$ for $NO_3^-$ and 3-6 µmol $L^{-1}$ for $NH_4^+$; Williams et al, 2009). Sources of nitrogen and sulfur compounds in the Arctic include long-range transport of fossil fuel combustion products from e.g. large smelters in the Russian Federation, shipping on Arctic sea routes, volcanic activity (e.g. Iceland, Alaska) and biomass combustion from natural or anthropogenic fires in the boreal forest zone (Hole et al., 2006b). The mix of Eurasian and North American sources for both S and N was found to be consistent across the Greenland ice sheet, based on emission inventories and ice core records (Fischer et al., 1998b), but it

may be assumed that local shipping sources would be greatest for the coastal region.

### 4.2 Deposition estimates

There are very few data for recent atmospheric deposition in Greenland, but there have been studies of snowpack and ice core records of pollutants on the Greenland ice sheet (e.g. Dye2, 200 km from Kangerlussuaq; Dye3, 380 km; Summit, 800 km; Burkhart et al., 2006). Therefore, despite the lack of contemporary deposition data for the region, there are numerous records of relative change in nitrogen deposition loads over the past 200 years or more which provide evidence of changes in long-range as opposed to local emission sources and assist in the interpretation of the spatial deposition patterns observed in the current study.

Ice-core records from Greenland show that increases in $NO_3^-$ commenced in the latter half of the nineteenth century (Mayewski et al., 1990; Fischer et al., 1998b; Burkhart et al., 2006). Greenland ice core records closely follow the emissions inventories for Europe and North America over this period (Burkhart et al., 2006). Current $NO_3^-$ concentrations are double the pre-industrial levels across the Greenland ice sheet (Fischer et al., 1998b; Hastings et al., 2009). Burkhart et al. (2006) calculated 1789-1994 deposition fluxes for six ice sheet cores to range from 0.13 to 0.59 kg ha$^{-1}$ a$^{-1}$ with mean N deposition flux derived from snow pits at Summit of 0.5 kg ha$^{-1}$ a$^{-1}$ as $NO_3^-$ (0.11 kgN ha$^{-1}$ a$^{-1}$; Burkhart et al., 2004). These fluxes are double the pre-industrial values for both wet and dry $NO_3^-$ deposition (Fischer et al., 1998a).

Fluxes of $NO_3^-$ deposition in the current study range from 0.08 to 0.13 kg ha$^{-1}$ a$^{-1}$ which are remarkably similar to the fluxes recorded at Summit by Burkhart et al. (2004). Total inorganic N deposition fluxes are 2-3 times higher at the coast (0.37 kg ha$^{-1}$ a$^{-1}$) than inland, primarily due to $NH_4^+$, reflecting both the higher precipitation but also possibly local sources, especially of $NH_4^+$. It is possible that there could be greater biogenic sources of $NH_4^+$ deposition from seabird colonies to the coastal sites which are within 2-3 km from the coast, but there are no recent data and older records indicate only very small colonies (20 pairs) within 10 km of the study region and the only sizeable colony of several hundred pairs is around 100 km away (Boertman et al., 1996). To our knowledge there are no major seabird colonies in the vicinity of the coastal sites and there is no evidence of elevated phosphate in the coastal snowpack which might be expected if seabirds were a major influence on snowpack chemistry. Unlike $NO_3^-$ there has been no similar increasing trend in $NH_4^+$ in ice core records over the past 200 years (Savarino & Legrand, 1998). Other studies of industrial sources of contaminants indicated by unsupported [210]Pb and weapons [137]Cs in lake sediment cores have also found a strong gradient of increasing deposition from the ice sheet to the coast in this region (Bindler et al., 2001a, b). Isotope ratios of Pb indicate that Western Europe is a major emissions source for southern Greenland (Bindler et al., 2001b).

Burkhart et al. (2004) reviewed several studies demonstrating that deposition flux (but not concentration) is strongly dependent on snow accumulation (Legrand & Kirchner, 1990), which is consistent with our results showing the much higher deposition flux at the coast where the snowpack is much greater than inland, even if only on a seasonal basis (unlike the ice

sheet). Unlike larger-scale studies of Arctic precipitation showing sulfur to be the main acidifying substance (Hole et al., 2009), $NO_3^-$ deposition appears to be of a similar magnitude in charge-equivalent terms to nss- $SO_4^{2-}$ deposition in inland regions of West Greenland, while TIN deposition is almost double nss-$SO_4^{2-}$ deposition (Table 5). At the coast, $NO_3^-$ deposition (9 mol ha$^{-1}$ a$^{-1}$) is much lower than nss-$SO_4^{2-}$ (22 mol ha$^{-1}$ a$^{-1}$ in charge-equivalent terms) while $NH_4^+$ is comparable (17 mol ha$^{-1}$ a$^{-1}$), possibly suggesting an influence of ammonium sulfate aerosols (Fisher et al., 2011; Paulot et al., 2015). Local urban, marine or shipping emissions could also account for higher deposition fluxes of all these ions in coastal snowpack, especially given the proximity of the town and port at Sisimiut, but since $NO_3^-$ fluxes are much less enhanced at the coast than either $NH_4^+$ or nss-$SO_4^{2-}$ a dominant deposition pathway via ammonium sulfate (Fisher et al., 2011) seems most likely. $NH_4^+$ and nss-$SO_4^{2-}$ are highly correlated in coastal snowpack (natural logs of concentrations; r=0.740, t=5.154, df=22, p=3.63e-05) which provides supporting evidence for this pathway.

Modelled wet deposition of nss-$SO_4^{2-}$ for West Greenland in a global analysis was found to be in the region of 0.2 kg ha$^{-1}$ a$^{-1}$ S (Vet et al., 2014), which falls in the middle of the spatial range suggested by our study (0.08 – 0.35 kg ha$^{-1}$ a$^{-1}$). The same modelling study indicated wet deposition of total N to be <1 kg ha$^{-1}$ a$^{-1}$, which includes the range of estimates in our study. Nitrogen deposition levels in West Greenland snowpack (assuming 50 % of deposition as snow) are comparable to those found in Svalbard snowpack, estimated at 0.059 kg ha$^{-1}$ a$^{-1}$ for $NO_3^-$-N and 0.03 kg ha$^{-1}$ a$^{-1}$ for $NH_4^+$-N in 2001 (Tye and Heaton, 2007) and $NO_3^-$-N deposition rates (wintertime, snow-derived only) to Alaskan snowpack of 0.07-0.12 kg ha$^{-1}$ a$^{-1}$ (Jaffe & Zukowski, 1993). These values are much lower than those found in high altitude snowpack in the Rockies of the USA, for example 1.43-1.71 and 0.46-0.81 kg ha$^{-1}$ a$^{-1}$ for $NO_3^-$-N and $NH_4^+$-N in the study of Williams et al (2009). Likewise, higher deposition loads are recorded in the Russian Arctic, where total N and S deposition loads range from 0.75-3.10 and 0.40-3.00 (much higher close to smelters) kg ha$^{-1}$ a$^{-1}$ (Hole et al., 2006a).

Finally, while our deposition estimates are comparable to modelled values and other Arctic studies in regions remote from pollution sources, the reliance of our estimates on chemical data from snowpack and a small number of rainfall samples means that our estimates should be viewed as approximate. Comparison of rainfall chemistry with snowpack within each region suggests that mean rainfall concentrations could be as low as 60 % or as high as 170 % of those in snowpack, with uncertainties therefore conservatively in the range ± 40 % for annual mean deposition fluxes.

### 4.3 Stable isotopes

Isotope delta values of snowpack $NO_3^-$ in the current study are comparable to the few other published studies in the Arctic from seasonal snowpack (as opposed to accumulating snow on the ice sheet). Heaton et al. (2004) sampling snowpack in Svalbard during 2001-2003 recorded $\delta(^{15}N)$ in the range -4 to -18 ‰ while $\delta(^{18}O)$ values fell in the range 42-76 ‰ (60-85 ‰ when accounting for organic contamination) while Tye and Heaton (2007), also in Svalbard, found seasonal snowpack

$\delta(^{15}N)$ fell in the range -7 to -18 ‰ while $\delta(^{18}O)$ values fell in the range 74-78 ‰. The snowpack data presented here are slightly higher but largely overlapping with the Svalbard studies, with the mean $\delta(^{15}N)$ of -11 ‰ for coastal catchments being lower than most of the non-polar studies reviewed by Heaton et al. (2004).

In the current study $\delta(^{15}N)$ is similar to seasonal snowpack (Heaton et al., 2004) and ice cores (Vega et al., 2015a) from Svalbard, but much lower (regional means from -5.7 to -11.3 ‰) than for $NO_3^-$ in ice cores obtained on the Greenland ice sheet, where $\delta(^{15}N)$ over the last 300 years declined from a pre-industrial value near +11 ‰ to values around -1 ‰ in the last decade (Hastings et al., 2009) and is closely correlated with fossil fuel emissions since 1750. While the decrease in $\delta(^{15}N)$ commenced around 1850, rising $NO_3^-$ mass fractions in snow from the pre-industrial value of 73 ng g$^{-1}$ only became apparent
later, from around 1890, reaching 133 ng g$^{-1}$ post 1950.

There are no published studies on the triple isotope analysis of O in coastal Greenland $NO_3^-$. Although $\delta(^{18}O)$ values in our study are in a similar range (81-84 ‰ cf. 60-95 ‰), values of $\Delta(^{17}O)$ for snowpack $NO_3^-$ are somewhat higher than those reported for atmospheric sources (aerosols, fog and precipitation) in a semiarid region of California (26 ±3 ‰; Michalski *et*
*al*, 2004). Michalski et al. (2003) showed in a study of seasonal isotopic composition that $\Delta(^{17}O)$  values were consistently higher in winter months. The high values of $\Delta(^{17}O)$ found here are comparable to other Arctic studies (Kunasek et al., 2008; Geng et al., 2015; Fibiger et al., 2016) and indeed such high values are only found in polar regions (Morin et al., 2009). Morin et al. (2007a) reported $\Delta(^{17}O)$  values of 29-35 ‰ at Alert, Canada and 26-36 ‰ at Barrow, Alaska, compared with the present study where the coastal mean value of 30.8 ‰ was significantly lower than both Kelly Ville (mean = 34.4 ‰)
and the ice sheet (mean = 33.8 ‰). However, some of the ice sheet margin samples in the current study ranged up to 43 ‰, higher than any previously recorded in the Arctic and comparable to data from Savarino et al. (2006) in Antarctica (maximum 43.1 ‰).

The non-mass dependent fraction of oxygen associated with tropospheric ozone means that there is positive correlation
between ozone concentration and $\Delta(^{17}O)$ (Morin et al., 2007b). In Arctic coastal zones, springtime ozone depletion events (ODEs) commonly occur due to reaction pathways involving marine derived halogen compounds and radicals, most importantly linked to bromine (Morin et al., 2007b). Morin et al. (2007b) established a significant positive correlation between $\Delta(^{17}O)$ and ozone concentration and it may be speculated that the lower $\Delta(^{17}O)$ values in coastal snowpack in our study could be linked to ODEs caused by marine influences; although bromide was not measured, the contribution of sea
salts to coastal snowpack in our study is very significantly greater than inland, suggesting a much greater potential for the influence of ODEs on $\Delta(^{17}O)$.

Hence there is a very strong gradient of declining snowpack $\delta(^{15}N)$ from the central ice sheet (Hastings et al., 2009), to the ice sheet margin and with the most depleted values at the coast, while $\Delta(^{17}O)$ is also lower at the coast. There is clearly a major difference in the $\delta(^{15}N)$ of continuously accumulating snow on the central Greenland ice sheet compared with seasonal snowpack in the zone from the ice sheet margin to the coast. Such a strong spatial gradient must reflect either:

    i.    differing isotopic composition of inputs, due to differing sources of snowpack $NO_3^-$ and/or fractionation during transport to the deposition site (cf. Morin et al., 2009; Vega et al., 2015b), or

    ii.    a gradient in post-depositional processing and fractionation of $NO_3^-$ between the coastal, inland and ice sheet sites.

### 4.3.1 Sources of snowpack $NO_3^-$

There are very few studies in West Greenland to provide evidence for the likely source regions for anthropogenic N or other acid deposition precursors. Kahl et al. (1997) argue that trajectories to Summit on the central ice sheet are similar to Dye 3 on the ice sheet in south Greenland (Davidson et al., 1993), and that in winter, 94% belong to westerly transport patterns (in fact moving from south-west coastal zones north-east onto the ice sheet). Geng et al. (2014) assume the dominance of North American pollutant sources at Summit. For our sites in West Greenland it appears that similar long-range source areas would

apply. Alternative approaches (lake sediment records of Pb isotopes) have indicated that European sources are also important contributors to pollution across the region (Bindler et al., 2001a, b), while the modelling study of Zatko et al. (2016) suggests that our study region is an area of wind convergence with air flow mainly from the interior down to the coast. Hence there is no clear indication in the literature of the key local source regions affecting our study areas, but some evidence that coastal and inland areas are likely to be exposed to similar long-range sources.

While it is possible there may be major differences in pollutant source regions across Greenland, in particular from the coast to the interior, the spatial scope of our study is very small relative to the size of the ice sheet and the modelled gradients shown by Zatko et al. (2016). Hence while differential source regions cannot be ruled out, the study areas are very close to each other relative to distances from source regions. A striking result is the similarity of the coastal isotopic data (both $\delta(^{15}N)$

and $\Delta(^{17}O)$ in seasonal snowpack but also in the summer rain samples) with studies much further afield including snowpack at Summit on the ice sheet, and atmospheric $NO_3^-$ at Alert, Canada and Barrow, Alaska (Hastings et al., 2004; Kunasek et al., 2008; Morin et al., 2012; Fibiger et al., 2016). Hence it seems unlikely that differences in source region are a plausible explanation for the spatial differences in the isotopic signatures of snowpack observed in our study. The modelling study of Zatko et al. (2016) also shows an increase in the proportional loss of $NO_3^-$ through photolysis moving inland from the coast

towards the ice sheet. While not directly applicable due to the lack of permanent snow cover in the present study transect, their modelled enrichment of ice-core $NO_3^-$ would be in the range 1-5 ‰ for the inland regions of our study and zero at the coast, which is entirely consistent with our findings in the seasonal snowpack.

The linkages between the $\delta(^{15}N)$ of $NO_3^-$ preserved in ice cores and anthropogenic sources are poorly understood and debated in the literature and similar questions arise in our study where coastal snowpack $\delta(^{15}N)$ is much more depleted than values reported for many emission sources. Fibiger and Hastings (2016) reviewed the published ranges of $\delta(^{15}N)$ of $NO_x$ and found that coal-fired power plant emissions were between +6 and +26 ‰, while only soil emissions and automobile emissions included values of less than -10 ‰. Their own data on experimental biomass burning indicated values from -7 to +12 ‰ with the majority of materials giving positive values, although black spruce found in northern latitudes did give the most negative values in their study. In a separate study of vehicle emissions which reported a wide range in $\delta(^{15}N)$ of $NO_x$ from -19.1 to 9.8 ‰, it was found that emissions from diesel powered vehicles were the lowest (Walters et al., 2015a). In our coastal sites, it is possible that $NO_x$ sources from diesel vehicles and shipping may contribute to snowpack $NO_3^-$, but while our coastal sites showed the lowest $\delta(^{15}N)$ values, other studies have found shipping emissions to be enriched in $^{15}N$ (Beyn et al., 2015). At the coast, the most $^{15}N$-enriched snowpack is found at AT5 closest to Sisimiut, a town with road vehicles, a port and an airport, while at Kelly Ville the most $^{15}N$-enriched snowpack was found at SS02, closest to the harbour and Kangerlussuaq (Fig. 1; Supplementary Information Figure S.I. 2). Hence while the influence of local sources cannot be ruled out, comparison of differences between catchments within regions does not support proximity to local sources as an explanation of the regional gradient in $\delta(^{15}N)$. An alternative hypothesis would be differences in source areas for long-range transported pollutants.

Heaton et al. (2004) speculate that preferential deposition of enriched $NO_3^-$ leads to increasingly depleted $NO_3^-$ with distance from source and the later study of Vega et al. (2015b) supported the presence of such a process in air masses travelling long distances over the Arctic. In the present study, it is possible that this process could account for the very low $\delta(^{15}N)$ found especially in coastal snowpack, but is unlikely to account for the regional gradient observed. Given the relatively small distances involved (of the order of 100 km from the ice sheet margin to the coast) relative to transport distances from possible N sources (assumed to be industrial regions in Europe, Siberia or North America ), the large difference in snowpack $\delta(^{15}N)$ (regional differences >5 ‰) seems unlikely to be caused exclusively by this process. Burkhart et al. (2006) speculate that the patterns of recent declining trends in ice-core $NO_3^-$ since the 1990s suggest that Greenland snow may be recording European and North American $NO_x$ and the distance of these sources from the study region are much greater than the within-region distances showing the gradient in isotopic composition.

Another possibility is that there could be a greater proportional contribution of dry deposition at the low precipitation inland sites, relative to the coastal sites. Studies of daily variations in surface snow chemistry and isotopic composition at a coastal site in Svalbard indicated that increasing $NO_3^-$ concentrations occurred between precipitation events, due to dry deposition inputs (Björkman et al., 2014). Since gas phase and aerosol $NO_3^-$ may be enriched in $^{15}N$ compared to wet deposited $NO_3^-$ (Heaton, 1987; Freyer, 1991; Garten, 1996; Elliott et al., 2009) such a mechanism could contribute to both the spatial patterns in $NO_3^-$ concentrations and isotopic differences observed in our study. While the Svalbard study of Björkman et al.

(2014) considered coastal snowpack and concluded that dry deposition processes were likely to be more important than post-depositional processing, our study regions cover a strong climatic gradient with a much greater potential role for sublimation and photolytic effects on snowpack $NO_3^-$ in inland sites.

### 4.3.2 Post-depositional processing

Higher levels of volatilisation of $NO_3^-$ at the inland sites with greater sublimation and lower precipitation (cf. 250 mm SWE at Summit; Dibb & Fahnestock, 2004) may lead to enrichment of snowpack $^{15}N$ compared with the coastal sites. Johansson et al. (2015) recorded a mean sublimation rate at Two-Boat lake (our ice sheet site SS903) of 0.63 mm d$^{-1}$ in April 2013 which is an order of magnitude greater than those recorded in the Svalbard study of Björkman et al. (2014) (0.042 mm d$^{-1}$) which ruled out post-depositional processing as a major determinant of snowpack $NO_3^-$ concentrations and isotopic composition. Heaton et al. (2004) and Morin et al. (2008) suggested that post-depositional processing of snowpack $NO_3^-$ would lead to isotopic enrichment, so while these processes cannot account for the low coastal values, they could account for the higher inland values if it is assumed that fresh snow in all regions started from a similar value. A fresh snow sample collected at Kelly Ville did indeed show a much lower $\delta(^{15}N)$ of -11.8 ‰ (Table 5) compared with total snowpack in the region, but ad hoc rainfall samples at different times of year showed a variable $\delta(^{15}N)$. Since snow photochemistry is a major driver of $NO_3^-$ re-emissions, the effects of post-depositional processing should be maximal in spring when UV exposure is highest and there is still snowpack present (Morin et al., 2008).

The observed spatial isotopic gradient could potentially be the result of two opposing processes which could act to produce the same gradient; higher melting losses at the coast and higher sublimation losses inland. At the coast, higher temperatures may result in greater melting and preferential elution of the heavier isotope, leaving a more depleted snowpack. Inland, lower temperatures reduce melting effects but lower cloud cover and precipitation along with a much smaller snowpack cause greater relative sublimation losses, leading to isotopic enrichment of the remaining snowpack. Such a process could explain the much more depleted $\delta(^{15}N)$ of -11.8 ‰ in fresh snow at Kelly Ville compared with a mean of -5.7 ‰ in total accumulated snowpack sampled at the same time of year but representing the net effect of post-depositional processing on the snowpack remaining at the end of the season.

While the relative importance of these processes cannot be determined conclusively from the current study, there are additional clues when comparing the terrestrial snowpack with the lake ice snowpack. At the coast and Kelly Ville, $NO_3^-$ concentrations are significantly higher in lake ice snowpack than in terrestrial snowpack (Fig. 3) and it may be speculated that this could be due to meltwater losses draining from catchment slopes (with elevated ionic concentrations due to preferential elution) accumulating on the frozen lake surface. However, $\delta(^{15}N)$ values are generally lower in lake ice snowpack than in the terrestrial snowpack (Fig S.I. 1) while the opposite might be expected if the lake ice snowpack was

receiving enriched meltwater from the catchment. Hence the most plausible mechanism which could decrease $NO_3^-$ concentrations on catchment slopes while increasing $\delta(^{15}N)$ would be greater volatilisation or sublimation losses of $NO_3^-$ to the atmosphere. It is possible that the snow accumulated on lake ice is of a different age mix than that sampled on catchment slopes, due to differential removal and redeposition during wind redistribution. Since higher concentrations of most ions were recorded in lake-ice snowpack (significant for $NO_3^-$ at the coast and Kellyville) it may be hypothesized that post-depositional losses of $NO_3^-$ are enhanced in snow on catchment slopes. Such a mechanism is also supported by the isotope data whereby terrestrial snowpack has generally higher $\delta(^{15}N)$ than lake ice snowpack, suggesting post-depositional enrichment. It is not possible to determine how snow has been redistributed in the current study (and in fact would be extremely difficult to measure in practice), but the consistent pattern for all lake catchments in all regions does suggest a common process operating across the study region.

Our data suggest that coastal snowpack more closely represents the source isotopic composition while increased post-depositional processing occurs moving inland as precipitation levels and snowpack accumulation rates decrease. Periodic melting events indicated by ice layers in the coastal snowpack may facilitate the downward transport of the relatively depleted $NO_3^-$, further reducing the potential for post-depositional processing via volatilisation or photolysis. The least depleted (or most enriched) $\delta(^{15}N)$ values in our study are found in the Kelly Ville region which has the lowest precipitation.

Burkhart et al. (2004) observed that almost all $NO_3^-$ found in surface snow at Summit was still present in firn snow pits one year later, while acknowledging that post-depositional $NO_3^-$ loss to the atmosphere may occur and can be offset by dry deposition of $HNO_3$. Concentrations of $SO_4^{2-}$ and $NO_3^-$ in firn are strongly affected by snow accumulation rates and this is particularly important for accumulation of $NO_3^-$ in snowpack since high $NO_3^-$ re-emission losses have been recorded in low accumulation areas such as central Antarctica (Fischer et al., 1998a). Likewise, Dibb et al. (2007) found $NO_3^-$ concentrations to be 9% higher in surface snow than in buried snow and concluded that post-depositional losses of $NO_3^-$ may be as high as 25% within 1-2 years of deposition. They attributed post-deposition losses of volatile species on ice grain surfaces to decreases in surface area/volume ratios due to ice grain growth, or to photolysis, while non-volatiles may increase due to either dry deposition and/or loss of water mass by sublimation. Post-deposition processing is likely to play a more significant role in areas of higher temperatures and/or lower accumulation rates (Burkhart et al., 2004; Fischer et al., 1998a).

Since post-depositional processing occurs primarily in the photic zone of the snowpack (modelled values from 6-51 cm in Greenland in the study of Zatko et al., 2016), a larger proportion of the snowpack at the inland sites must be exposed to such processing during spring, while much deeper snowpack at the coast will retain a greater proportion of unprocessed $NO_3^-$. Although dust inputs are likely to be greater at the inland sites, potentially reducing the depth of the snow photic zone, the much smaller snowpack and greater wind redistribution suggests a much greater potential overall for post-depositional processing through UV exposure and wind removal of photolysis or evaporative products than at the coast (cf. Frey et al.,

2009). Frey et al. (2005) found that at wind speeds of less than 3 m s$^{-1}$ (as found at our coastal sites) the effects of wind-pumping were less important than diffusion; while our inland regions experience higher mean annual wind speeds of 3.6 (Kellyville) and 4.0 (ice sheet) m s$^{-1}$. Furthermore, several studies of both modern snowpack and ice core NO$_3^-$ (e.g. Geng et al., 2015) attribute differences in NO$_3^-$ $\delta(^{15}$N) to differences in snow accumulation rate, which is consistent with results of our study showing a less-transformed snowpack NO$_3^-$ signal at the coast. Frey et al. (2009) also highlighted the importance of surface and wind-driven sublimation processes in the enrichment of insoluble chemical species and the removal of volatile species. Their study, like ours, indicated smaller NO$_3^-$ transformations from snowpack in higher accumulating areas at the coast compared with inland, and the analysis of Zatko et al (2016) in both Antarctica and on the Greenland ice sheet found that enrichment of snowpack NO$_3^-$ was greatest in areas with the lowest accumulation rates – consistent with our data from seasonal snowpack.

The modelling study of Zatko et al. (2016) also indicates that up to 100% of snowpack NO$_3^-$ deposition in West Greenland is primary deposition, rather than recycled. Our data, if we assume that coastal snowpack NO$_3^-$ most closely represents regionally deposited precipitation NO$_3^-$, indicate an enrichment in $\delta(^{15}$N) of 3.8 ‰ at the ice sheet and 5.6 ‰ at Kellyville, while $\Delta(^{17}$O) is 3.0 ‰ higher at the ice sheet and 3.6 ‰ higher at Kellyville, relative to coastal snowpack. The lack of a concomitant decrease in $\delta(^{18}$O) for inland snowpack suggests the post-depositional enrichment in $\delta(^{15}$N) may be due primarily to net losses from snowpack rather than in-situ recycling. Slightly higher mean values of $\delta(^{18}$O) at inland locations, while not significant, are also suggestive of fractionating losses, rather than in situ recycling which would be expected to reduce $\delta(^{18}$O). Given that $\delta(^{15}$N) shows an increase without a concomitant decrease in $\Delta(^{17}$O), NO$_3^-$ loss rather than recycling would appear to be the dominant process at inland sites, which is consistent with the presence of a much smaller, more sublimated and wind-redistributed snowpack inland which favours removal and transport of photolytic and evaporative products rather than in situ recycling.

If the much higher $\delta(^{15}$N) values inland do indeed reflect a much greater impact of post-depositional processing on the much smaller snowpack, then it follows that the initial snowpack deposition of NO$_3^-$ may have been larger, but has subsequently been reduced by photolysis and evaporation, while coastal snowpack more faithfully records the initial atmospheric inputs of NO$_3^-$. For the purpose of calculating net deposition rates to catchments and receiving lake basins, the net effects (photochemical losses and gains) on NO$_3^-$ through post-depositional processing throughout the snow accumulation season should be accounted for by sampling at the end of the season.

### 4.3.3 Seasonal variations in $\delta(^{15}$N) of deposited NO$_3^-$ versus annually integrated ice core records

Ice core data show seasonal variation in the recent isotopic signature, with summer values higher than winter, which was not apparent in pre-industrial ice (Hastings et al., 2004, 2009). The $^{15}$N depletion is strongest in winter. Since our study records

only winter deposition inputs, it is likely that we are capturing the most depleted component of annual inputs leading to lower values than annually resolved records on the ice sheet, where snowpack continuously accumulates through the year (Dibb & Fahnestock, 2004). Morin et al. (2008) compared their coastal snowpack data with the strong seasonal variations in atmospheric $\delta(^{15}N)$ on the ice sheet, with lowest values of -15 ‰ in winter through to March, and asserted that emissions of reactive N from snowpack during spring resulted in an increase of $\delta(^{15}N)$ in remaining $NO_3^-$. Hastings et al. (2004) found a strong seasonal variation in the $\delta(^{15}N)$ of fresh snow, with minimal mean values in winter of -10 ‰, and even recorded diurnal variations in deposited snow, with $^{15}N$-enriched snow samples collected during the day and $^{15}N$-depleted ones at night. They attributed this diurnal signal to redeposition of $NO_x$ emitted from the snowpack during the day, either through direct contact with the snow surface or during fog events. These patterns are consistent with the spatial gradient in the current study, where the most depleted snowpack $\delta(^{15}N)$ was found at the coast, where higher precipitation (and accumulation) and greater incidence of cloud cover and fog would reduce the potential for re-emission of $^{15}N$-depleted N from the snowpack. The few rainfall samples analysed during late summer for all regions show a higher $\delta(^{15}N)$ than the snowpack samples, while the sole rain sample analysed from spring (May 2011) at the ice sheet margin had the lowest value in rain, but still not as low as snowpack. Such post-depositional processing in combination with seasonal changes found on the ice sheet, and hinted at in our rainfall data, could explain the major differences between our coastal snowpack samples (reflecting minimal winter deposition) and those recorded at Summit (reflecting year round accumulation including much more enriched summer deposition).

## 5. Conclusions

There are major differences in snowpack accumulation and SWE from inland to the coast, reflecting the annual precipitation which is twice as high at the coast than inland. Late season snowpack in West Greenland shows a strong chemical gradient from the ice sheet margin to the coast. For inland snowpack, chemistry is comparable to remote locations such as Summit on the central ice sheet as well as other Arctic locations remote from industrial sources. At the coast, sea salt ions dominate the accumulated snowpack, but are much less important in fresh snow. While $NO_3^-$ is the dominant ion at Summit (Dibb et al., 2007) its concentration declines from inland regions to the coast. However the reverse is true for $NH_4^+$ and nss-$SO_4^{2-}$ with significantly higher concentrations in coastal snowpack than inland. Marine-derived aerosols of ammonium sulfate may be a possible source of these ions in coastal snowpack.

A lack of summer rainfall chemistry data prevents accurate estimation of annual deposition fluxes, but net deposition inputs to catchments may be approximated by assuming (on the basis of a small number of *ad hoc* rainfall samples) that snowpack chemistry is representative of annual mean precipitation, since snow represents around half of annual precipitation. On this assumption there is a strong deposition gradient from inland to the coast, which is much more pronounced for $NH_4^+$ and nss-

$SO_4^{2-}$ than for $NO_3^-$. While $NO_3^-$ deposition ranges from 0.08-0.13 kg ha$^{-1}$ a$^{-1}$, comparable to fluxes at Summit (Burkhart et al., 2004), total inorganic N deposition fluxes are almost 2-3 times higher at the coast (0.37 kg ha$^{-1}$ a$^{-1}$) than inland, primarily due to $NH_4^+$. These values are within the range of other studies from remote Arctic locations. In charge equivalent terms, $NO_3^-$ deposition is very similar to nss-$SO_4^{2-}$ at inland locations, but less than half at the coast.

While chemistry and deposition show similarities to other studies of ice sheet snowpack, stable isotope data show major differences. There is a gradient of declining $\delta(^{15}N)$ from the inland areas to the coast, but samples are all $^{15}$N-depleted compared with samples from Summit, with regional means from -5.7 to -11.3 ‰ compared with only -1 ‰ in recent Summit ice core samples. While differences in emissions sources are possible, post-depositional processing of snowpack $NO_3^-$ seems

the most plausible mechanism driving this very strong gradient in N fractionation. The processes that best explain both the spatial gradient and also the observed differences between snowpack on catchment slopes and lake ice snowpack are sublimation and volatilization. These processes can act to reduce $NO_3^-$ concentrations while simultaneously increasing the remaining snowpack $\delta(^{15}N)$. Hence lower $NO_3^-$ concentrations in coastal snowpack are due to the diluting effect of much higher precipitation, but lower concentrations on catchment slopes within each region compared with lake ice snowpack are

due to enhanced losses to the atmosphere. We conclude that at our inland regions, but especially at Kelly Ville, lower precipitation and snowpack accumulation in combination with higher wind speeds enhances both photolytic and physical (sublimation & evaporative) losses of snowpack $NO_3^-$. Since we see a significant enrichment in $\delta(^{15}N)$ but not in $\delta(^{18}O)$ (inland mean values are higher, but not significantly different from the coast) we suggest that in situ recycling is less important than net losses through photolysis and wind removal of $NO_x$. Physical losses would also lead to $\delta(^{15}N)$ and $\delta(^{18}O)$

enrichment of remaining snowpack $NO_3^-$ without affecting $\Delta(^{17}O)$. However, we find significantly higher $\Delta(^{17}O)$ at our inland sites compared with coastal sites, in combination with higher $\delta(^{15}N)$, but no difference in $\delta(^{18}O)$. The higher $\Delta(^{17}O)$ found in the inland regions must therefore reflect a greater role for the $O_3$ oxidation pathway as the source of snowpack $NO_3^-$, compared with the coastal sites. Overall, the isotopic composition of coastal snowpack reflects the low $\delta(^{15}N)$ of winter deposition observed in other studies at Summit and Svalbard, but the current study demonstrates for the first time the spatial

differences in snowpack isotopes resulting from the climatic gradient from the coast to the ice sheet.

Future changes in climate are likely to affect the gradients in snowpack chemistry, stable isotopes and deposition observed in the current study, given the importance of precipitation and other climatic factors in driving spatial differences. In 2012 the coastal town of Sisimiut recorded its highest annual precipitation since records began (1004 mm) and this AWS station

shows the strongest increasing precipitation trend across the Greenland network of +48.5 mm a$^{-2}$ from 2001-2012 (Mernild et al., 2015). Over the same period, Kangerlussuaq and the closest ice sheet sites showed decreasing trends in precipitation. It has also been speculated that climate change will affect the relative importance of source types and regions for deposition in the Arctic. While some of these sources are closely linked to industrial activity in the Arctic regions of Europe and the Russian Federation, emissions from forest fires and shipping activity could both increase with climatic warming in the Arctic

polar region (Hole et al., 2006b). On the basis of the current study, future changes in both the magnitude and isotopic signature of net anthropogenic nitrogen deposition delivered to Arctic ecosystems will be determined to a large degree by the effects of climatic change on spatial patterns of precipitation, snowpack accumulation and sublimation.

## Author contributions

C. J. Curtis, N.J. Anderson, G. Simpson, V. Jones and J. Kaiser designed the study. Curtis, Anderson, Simpson, Jones and Whiteford carried out fieldwork and sample preparation. Kaiser and Marca carried out laboratory analyses. All authors contributed to writing and interpretation of the data.

## Competing interests

The authors declare that they have no conflict of interest.

## Data availability

All underlying site location, chemistry and isotope data are publicly available via the following link: (DOI to follow)

## Acknowledgments

This project was funded by NERC (Long range atmospheric nitrogen deposition as a driver of ecological change in Arctic lakes; grant NE/G020027/1 to UCL, NE/G019509/1 to UEA and NE/G019622/1 to Loughborough University). We thank Karen Schleiss of Kangerlussuaq and Prof Morten Nielsen of DTU (Sisimiut) for ad hoc rainfall sampling. Figure 1 was produced by Wendy Phillips of GAES. The study would not have been possible without the dedication in the field from James Shilland, Simon Patrick, Ewan Shilland and Simon Turner of UCL.

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

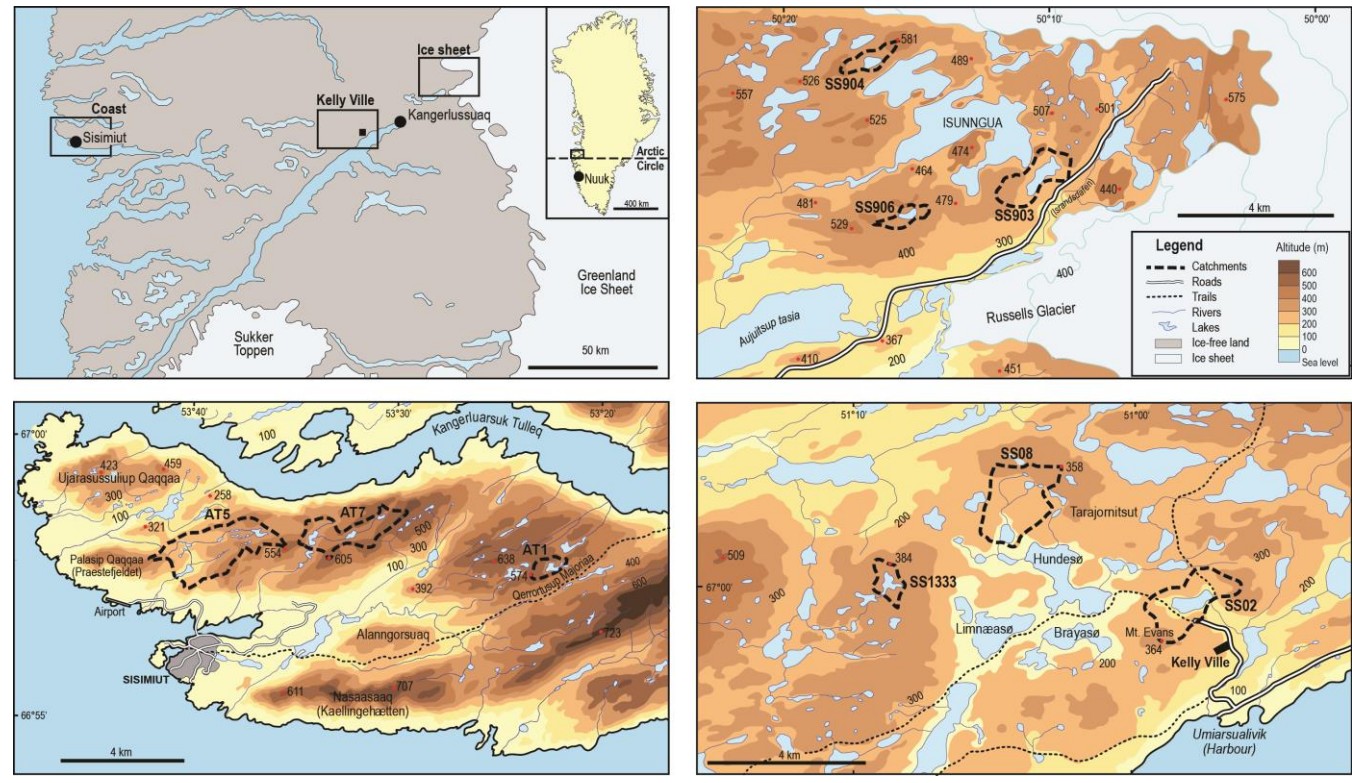

**Figure 1: Location of sampling regions and catchments within West Greenland**

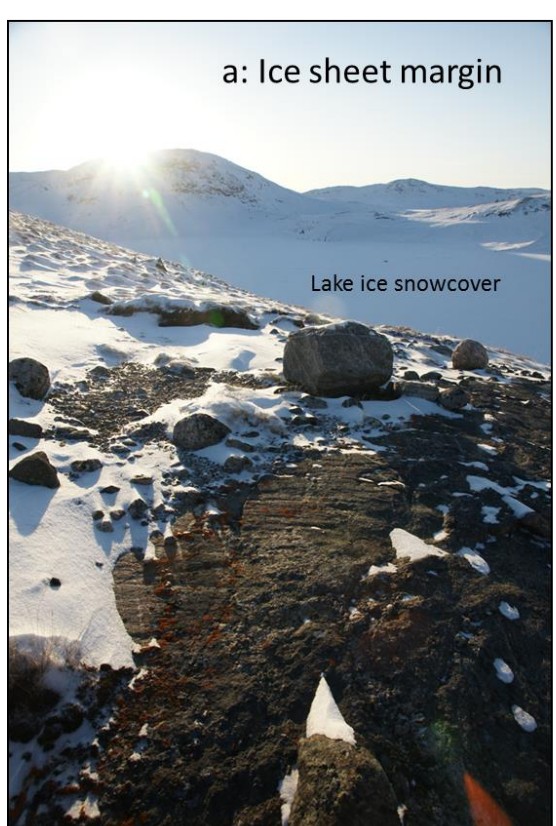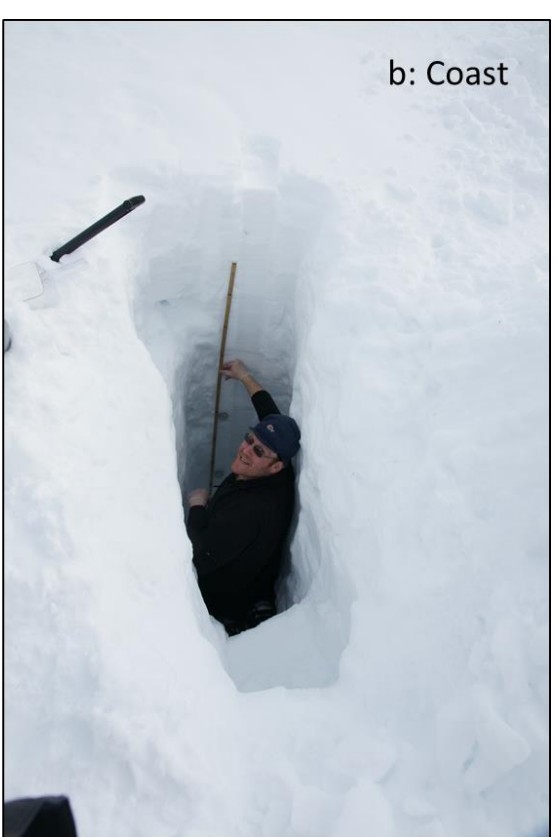

**Figure 2: a) Typical sublimating snowpack close to the ice sheet and lake ice snowcover on lake SS903 (24ᵗʰ March 2011), b) contrasting with deep snow profile on lower catchment slope of coastal site AT7 two days later (26ᵗʰ March 2011)**

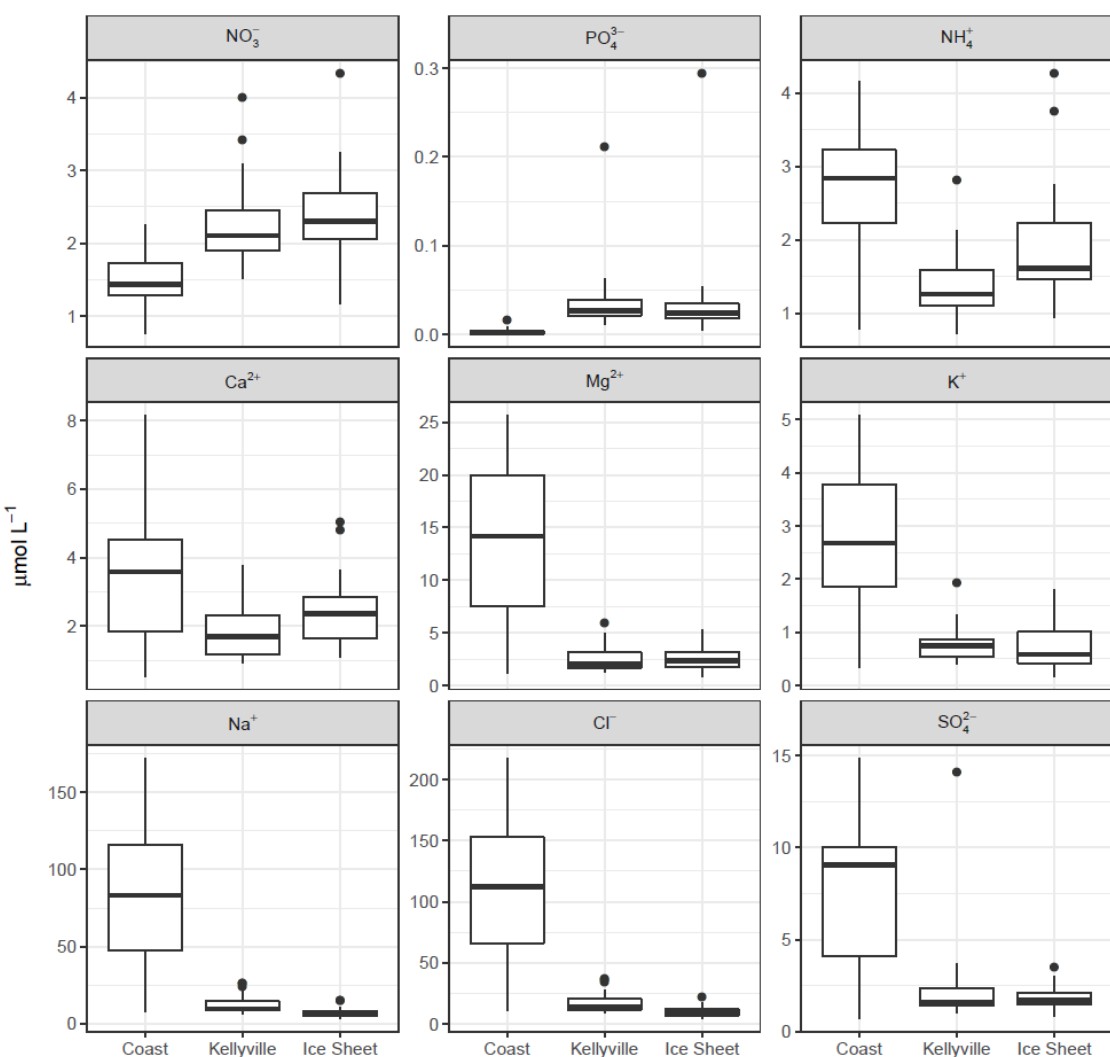

**Figure 3: Comparison of regional snowpack ion concentrations (aggregated data – n=24 per region; all units in µmol L$^{-1}$). Boxes represents 25$^{th}$ and 75$^{th}$ percentiles, horizontal line = median, whiskers show data extent, points indicate outliers (1.5-3 IQRs outside box)**

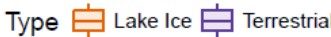

Figure 4: Comparison of lake ice and terrestrial snowpack concentrations (µmol L⁻¹) for selected ions. See Fig. 3 caption for explanation of boxplots.

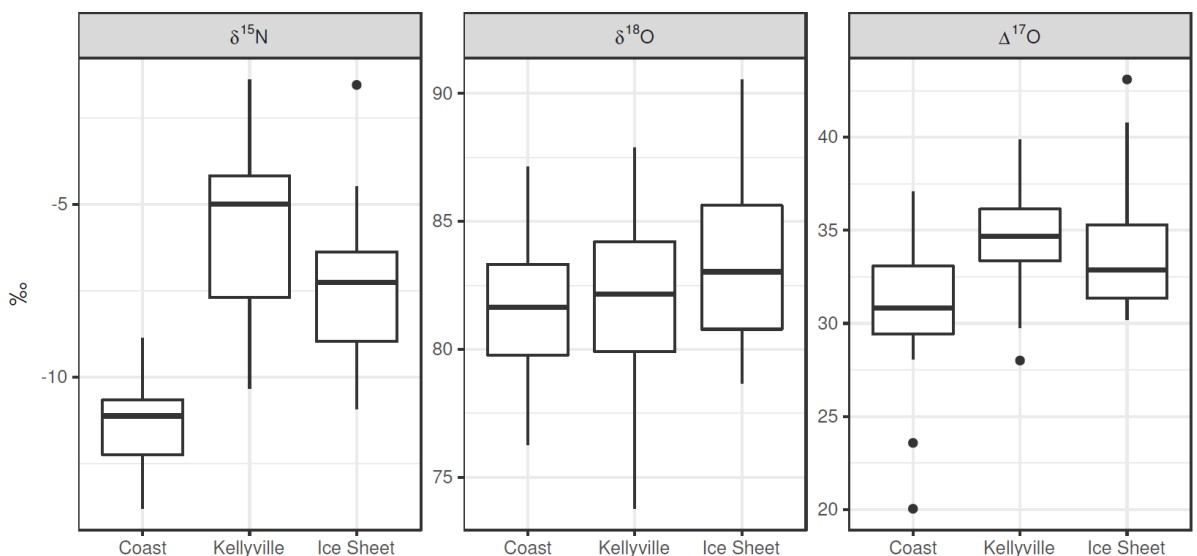

**Figure 5: Comparison of regional aggregated (n=24 per region) snowpack stable isotope composition of NO$_3^-$. See Fig.3 caption for explanation of boxplots.**

**Table 1: Sampling catchment details (based on centroid of lakes)**

| Region | Site | Latitude (° N) | Longitude (° W) | Altitude (m) | Lake area (ha) | Catchment area (ha) |
|--------|------|------|------|------|------|------|
| Ice Sheet | SS906 | 67.120 | 50.256 | 415 | 9.3 | 50.6 |
| Ice Sheet | SS903 | 67.130 | 50.172 | 315 | 38.0 | 117 |
| Ice Sheet | SS904 | 67.157 | 50.278 | 405 | 12.4 | 42.5 |
| Kelly Ville | SS02 | 66.996 | 50.964 | 160 | 36.8 | 217 |
| Kelly Ville | SS08 | 67.013 | 51.075 | 163 | 14.6 | 278 |
| Kelly Ville | SS1333 | 67.001 | 51.146 | 308 | 13.8 | 57.5 |
| Coast | AT1 | 66.967 | 53.404 | 445 | 8.0 | 49.7 |
| Coast | AT7 | 66.972 | 53.585 | 324 | 6.5 | 203 |
| Coast | AT5 | 66.961 | 53.679 | 117 | 7.5 | 443 |

5  **Table 2: Snow depth measurements and snow water equivalents (SWE) by catchment and region**

| Site / Region | Area (ha) | Altitude (m) Lake/Min. | Max. | *n* | Snow depth ( cm) Range | Mean | SD | *n* | SWE (mm) Mean | SD | Corr. SWE (mm) * |
|--------|------|------|------|------|------|------|------|------|------|------|------|
| SS906 | 50.6 | 415 | 463 | *64* | 1-55 | 20.1 | 13.9 | *48* | 37.9 | 35.9 | 34.5 |
| SS903 | 117 | 315 | 476 | *155* | 0-74 | 20.0 | 13.9 | *58* | 43.9 | 45.8 | 38.6 |
| SS904 | 42.5 | 405 | 487 | *104* | 0-103 | 16.1 | 17.5 | *26* | 49.9 | 54.9 | 35.1 |
| **Ice Sheet** | | **315** | **487** | ***323*** | **0-103** | **18.8** | **15.2** | ***132*** | **42.9** | **44.4** | **35.7** |
| SS02 | 217 | 160 | 372 | *123* | 0-65 | 20.5 | 12.0 | *65* | 39.1 | 27.7 | 35.3 |
| SS08 | 278 | 163 | 279 | *120* | 2-50 | 23.8 | 10.7 | *49* | 49.6 | 33.9 | 48.8 |
| SS1333 | 57.5 | 308 | 355 | *94* | 1-100 | 24.4 | 15.3 | *39* | 53.3 | 41.7 | 48.6 |
| **Kelly Ville** | | **160** | **372** | ***337*** | **0-100** | **22.8** | **12.7** | ***153*** | **46.1** | **34.0** | **43.6** |
| AT1 | 49.7 | 445 | 558 | *102* | 0-240 | 73.5 | 52.8 | *48* | 187.9 | 119.3 | 196.1 |
| AT7 | 203 | 324 | 441 | *111* | 1-260 | 49.7 | 43.6 | *48* | 149.7 | 93.6 | 161.3 |
| AT5 | 443 | 117 | 197 | *112* | 5-250 | 66.9 | 47.0 | *45* | 135.6 | 89.9 | 181.8 |
| **Coast** | | **117** | **551** | ***325*** | **0-260** | **63.1** | **48.7** | ***141*** | **158.0** | **103.5** | **180.8** |

*****Mean SWE corrected for larger number of snow depth-only measurements**

**Table 3a: Catchment terrestrial snowpack sampling details (n=5 per catchment)**

| Region | Site | Date | Altitude (m) | Depth (cm) | Air temp (°C) | Snow temp (°C) |
|---|---|---|---|---|---|---|
| Ice Sheet | SS906 | 23/03/2011 | 430-440 | 24-34 | -18 to -22 | -14 to -27 |
| Ice Sheet | SS903 | 24/03/2011 | 333-371 | 27-74 | -10 to -18 | -6 to -20 |
| Ice Sheet | SS904 | 30/03/2011 | 421-450 | 25-103 | -6 to -11.5 | -4 to -8.5 |
| Kelly Ville | SS02 | 22/03/2011 | 192-260 | 23-65 | -16 to -25 | -13 to -31 |
| Kelly Ville | SS08 | 31/03/2011 | 198-211 | 40-48 | -8 to -11 | -8.5 to -13.5 |
| Kelly Ville | SS1333 | 01/04/2011 | 317-349 | 27-49 | -9 to -15 | -7 to -11 |
| Coast | AT1 | 27/03/2011 | 478-558 | 28-240 | 0 to 2 | -2.5 to -9 |
| Coast | AT7 | 26/03/2011 | 346-362 | 31-238 | 4 to 6 | -0.5 to -9 |
| Coast | AT5 | 28/03/2011 | 132-143 | 74-141 | -1 to 0.5 | -1.5 to -7 |

**Table 3b: Lake ice snowpack sampling details (n=3 per lake)**

| Region | Site | Date | Altitude (m) | Depth (cm) | Air temp (°C) | Snow temp (°C) |
|---|---|---|---|---|---|---|
| Ice Sheet | SS906 | 23/03/2011 | 421-430 | 12-23 | -18 | -10 to -18 |
| Ice Sheet | SS903 | 24/03/2011 | 333-340 | 8-13 | -12 to -18 | -11 to -15 |
| Ice Sheet | SS904 | 30/03/2011 | 414-430 | 5-10 | -11 to -12 | -5 to -8 |
| Kelly Ville | SS02 | 02/04/2011 | 169-180 | 15-23 | -9 to -12 | -6 to -9 |
| Kelly Ville | SS08 | 31/03/2011 | 166-178 | 20-30 | -8 to -11 | -6 to -10.5 |
| Kelly Ville | SS1333 | 01/04/2011 | 311-323 | 20-25 | -12 to -15 | -7 to -15 |
| Coast | AT1 | 27/03/2011 | 456-463 | 35-44 | 4 to 5 | -0.5 to -2 |
| Coast | AT7 | 26/03/2011 | 336-344 | 19-32 | 1 to 5 | -0.5 to -5 |
| Coast | AT5 | 28/03/2011 | 128-133 | 25-60 | 0 to 0.5 | -1 to -3 |

**Table 4: Comparison of aggregated snowpack chemistry (n=24 per region), derived non-marine concentrations of major ions (all in µmol L$^{-1}$) and isotopic composition (‰). See text for details of post-hoc pairwise comparisons. Coast = CO, Kelly Ville = KV, ice sheet margin = IS.**

| Region: | Coast (CO) | | Kelly Ville (KV) | | Ice Sheet (IS) | | Post hoc sig. differences | | |
| Analyte | Mean | SD | Mean | SD | Mean | SD | CO-KV | CO-IS | KV-IS |
|---|---|---|---|---|---|---|---|---|---|
| **Nutrients** | | | | | | | | | |
| NO$_2^-$ | **0.0** | 0.0 | **0.0** | 0.0 | **0.1** | 0.4 | | | |
| NH$_4^+$ | **2.7** | 0.9 | **1.4** | 0.5 | **1.9** | 0.8 | 0.0017 | | |
| NO$_3^-$ | **1.5** | 0.4 | **2.3** | 0.6 | **2.4** | 0.6 | <0.0001 | <0.0001 | |
| PO$_4^{3-}$ | **0.0** | 0.0 | **0.1** | 0.1 | **0.1** | 0.2 | <1e-05 | <1e-05 | |
| **Base cations** | | | | | | | | | |
| Ca$^{2+}$ | **3.5** | 2.0 | **1.8** | 0.8 | **2.5** | 1.0 | <0.001 | | |
| Mg$^{2+}$ | **13.8** | 7.4 | **2.6** | 1.2 | **2.5** | 1.1 | <1e-08 | <1e-08 | |
| K$^+$ | **2.8** | 1.4 | **0.8** | 0.3 | **0.8** | 0.5 | <1e-05 | <1e-05 | |
| Na$^+$ | **86.9** | 48.5 | **12.2** | 6.0 | **7.1** | 3.1 | <0.001 | <0.001 | |
| **Anions** | | | | | | | | | |
| Cl$^-$ | **114.4** | 61.0 | **17.2** | 8.6 | **10.3** | 4.3 | <0.0001 | <0.0001 | |
| SO$_4^{2-}$ | **7.7** | 4.2 | **2.4** | 2.6 | **1.8** | 0.6 | <0.0001 | <0.0001 | |
| **Non-sea salt** | | | | | | | | | |
| nss-Ca$^{2+}$ | **1.4** | 1.4 | **1.6** | 0.7 | **2.3** | 1.0 | <1e-08 | <1e-08 | <1e-08 |
| nss-Mg$^{2+}$ | **2.4** | 1.8 | **0.9** | 0.5 | **1.5** | 0.9 | <0.001 | | 0.014 |
| nss-K$^+$ | **0.7** | 0.4 | **0.5** | 0.3 | **0.6** | 0.4 | | | |
| nss-Na$^+$ | **-11.5** | 4.5 | **-2.5** | 1.3 | **-1.8** | 1.1 | | | |
| nss-SO$_4^{2-}$ | **1.8** | 1.8 | **1.0** | 0.4 | **1.2** | 0.5 | 0.0011 | 0.0358 | |
| **Isotopes** | | | | | | | | | |
| $\delta(^{15}N)$ | **-11.3** | 1.3 | **-5.7** | 2.3 | **-7.5** | 2.2 | <0.0001 | 0.001 | |
| $\delta(^{18}O)$ | **81.7** | 2.8 | **81.9** | 3.5 | **83.3** | 2.9 | | | |
| $\Delta(^{17}O)$ | **30.8** | 3.7 | **34.4** | 2.8 | **33.8** | 3.3 | <0.001 | 0.0052 | |

**Table 5: Comparison of *ad hoc* rain and fresh snow samples with mean snowpack data (from Table 4 - italics) (concentrations in µmol L$^{-1}$; isotopes in ‰)**

| Location | Sample type | Sample dates | NH$_4^+$ | NO$_3^-$ | PO$_4^{3-}$ | Ca$^{2+}$ | K$^+$ | Mg$^{2+}$ | Na$^+$ | Cl$^-$ | SO$_4^{2-}$ | $\delta(^{15}N)$ | $\delta(^{18}O)$ | $\Delta(^{17}O)$ |
|---|---|---|---|---|---|---|---|---|---|---|---|---|---|---|
| SS903 | Rain | 18/05/11-23/05/11 | 3.6 | 5.0 | 0.2 | 6.2 | 0.2 | 1.8 | 7.8 | 9.3 | 3.0 | -10.0 | 81.9 | 30.3 |
| SS906 | Rain | 05/09/12-06/09/12 | 2.6 | 1.7 | 0.2 | 9.0 | 2.6 | 2.8 | 13.8 | 18.3 | 2.0 | -5.6 | 60.8 | 20.6 |
| **Ice Sheet** | **Rain** | **Mean** | **3.1** | **3.4** | **0.2** | **7.6** | **1.4** | **2.3** | **10.8** | **13.8** | **2.5** | **-7.8** | **71.4** | **25.5** |
| *Ice Sheet* | *Snowpack* | *Mean* | *1.9* | *2.4* | *0.3* | *2.5* | *0.8* | *2.6* | *7.1* | *10.3* | *1.8* | *-7.5* | *83.3* | *33.8* |
| SS02 | Rain | 08/08/11-24/08/11 | 1.4 | 7.8 | 0.1 | 12.3 | 0.6 | 2.1 | 14.1 | 16.8 | 4.1 | 0.9 | 70.9 | 22.7 |
| SS02 | Rain | 28/08/11-01/09/11 | 1.3 | 5.2 | 0.0 | 7.6 | 3.5 | 0.7 | 5.7 | 4.1 | 1.5 | -0.9 | 76.4 | 23.1 |
| SS02 | Rain | 01/09/11-07/09/11 | 0.3 | 1.8 | 0.1 | 6.2 | 0.3 | 0.8 | 4.2 | 3.8 | 1.4 | -3.4 | 63.3 | 21.9 |
| Kanger. | Rain | 05/09/12-06/09/12 | 0.2 | 1.3 | 0.1 | 15.0 | 0.5 | 2.4 | 9.3 | 10.2 | 1.9 | 0.4 | 55.4 | - |
| **Kelly Ville** | **Rain** | **Mean** | **0.8** | **4.0** | **0.1** | **10.3** | **1.2** | **1.5** | **8.3** | **8.7** | **2.2** | **-0.8** | **66.5** | **22.6** |
| SS1333 | Snow Fresh | 01/04/11 | 3.4 | 3.4 | 0.3 | 1.9 | 0.9 | 2.7 | 13.5 | 18.8 | 2.6 | -11.8 | 84.2 | 31.4 |
| *Kelly Ville* | *Snowpack* | *Mean* | *1.4* | *2.3* | *0.3* | *1.9* | *0.8* | *2.6* | *12.2* | *17.2* | *2.4* | *-5.7* | *81.9* | *34.4* |
| AT5 | Snow Fresh | 28/03/11 | 4.6 | 4.7 | 0.0 | 1.2 | 0.5 | 2.2 | 10.6 | 16.3 | 2.3 | -10.2 | 83.6 | 31.2 |
| *Coast* | *Snowpack* | *Mean* | *2.7* | *1.5* | *0.0* | *3.5* | *2.8* | *13.8* | *86.9* | *114.4* | *7.7* | *-11.3* | *81.7* | *30.8* |

**Table 6: Mean annual meteorological data and deposition estimates based on snowpack chemistry and mean precipitation (2001-12) for Sisimiut (coast), Kangerlussuaq and SS903 at the ice sheet margin (Johansson et al., 2015; Mernild et al., 2015). #Data from Whiteford et al. (2016) are 2011 mean values except Kangerlussuaq (Kelly Ville) wind speed, which is from 2010**

| Region | Pptn | Temp[#] | Wind speed[#] | N in NO$_3^-$ | N in NH$_4^+$ | TIN | S in nss-SO$_4^{2-}$ | NO$_3^-$ | NH$_4^+$ | TIN | nss-SO$_4^{2-}$ |
|---|---|---|---|---|---|---|---|---|---|---|---|
| | mm | °C | m s$^{-1}$ | kg ha$^{-1}$ a$^{-1}$ | | | | mol ha$^{-1}$ a$^{-1}$ | | | |
| Coast | 631 | -2.1 | 2.9 | 0.13 | 0.24 | 0.37 | 0.35 | 9 | 17 | 27 | 11 |
| Kelly Ville | 258 | -5.6 | 3.6 | 0.08 | 0.05 | 0.13 | 0.08 | 6 | 4 | 10 | 3 |
| Ice sheet | 320 | -5.0 | 4.0 | 0.11 | 0.09 | 0.19 | 0.13 | 8 | 6 | 14 | 4 |

