# Peer review of "Spatial variations in snowpack chemistry, isotopic composition of NO3- and nitrogen deposition from the ice sheet margin to the coast in West Greenland"

_Biogeosciences, 2017_

## Referee Comment (RC1) · Anonymous Referee #1 · 16 Jun 2017

Curtis et al report measurements of ion concentrations and nitrate isotopes in 3 locations representing different snow accumulation regimes in western Greenland. All observations show gradients from the coast to the inland site on the ice sheet, with sea salt and sulfate concentrations highest at the coast while nitrate concentrations are highest inland. Most of their discussion focuses on nitrate and its nitrogen isotopic composition, where they conclude that postdepositional processing likely determines the observed spatial gradient. Given that the latter has been somewhat contested in the literature, such a study is important. They also provide estimates of the deposition flux of nitrate, ammonia, and sulfate at each location. The authors otherwise do not do as much analysis of the other data sets, such as the ions other than nitrate and oxygen

isotopic composition of nitrate.

Although the manuscript is well written as far as English language and grammar, it's missing some important background information making it somewhat hard to follow the analysis of the data. Some specific comments on this are below. The technical details seem scientifically sound. Abstract: The authors should start the abstract with a motivation for this study. Why should one be interested in the observed spatial gradients?

Introduction: The introduction needs more background information. It is very short relative to the length of the entire paper. The introduction should present the potential sources of the observed ions in Greenland and discuss what controls the isotopic composition of nitrate. It should include a discussion of postdepositional processing, which is never really defined. It should explicitly discuss why one should care about the observed spatial gradients, which seems to be the main motivation of the study.

Methods: Please state over what snow depth the snow samples were collected. Over the first 10 cm? Deeper? Shallower?

Figure 1: What do the colors mean?

Section 4.3.1: Provide a reference for the statement that "gas-phase aerosol NO3- may be enriched in 15N compared to wet deposited NO3-". Also, "gas-phase aerosol NO3-" does not make sense. Nitrate is either the gas-phase or the aerosol phase (i.e., equilibrium partitioning between the two phases).

Section 4.3.2: This section was particularly hard to read because postdepositional processing is never defined. Many studies on ice sheets have shown that photolysis dominates postdepositional processing, but this is not even mentioned until the very end of this section. Perhaps if the authors properly introduce this process in the introduction, it will make it easier to clarify this section as well. It would be useful to give the fractionation factors for the processes involved.

Conclusion: Like the abstract, the conclusion focuses on the observed gradients without explicitly stating why this matters. Again, a more thorough introduction may help with this.

Some relevant references that could be included in the introduction and/or discussion and data comparison:

Kunasek, S.A., Alexander, B., E.J. Steig, M.G. Hastings, D.J. Gleason and J.C. Jarvis, Measurements and modeling of $\Delta17O$ of nitrate in snowpits from Summit, Greenland, J. Geophys. Res., 113, D24302 (2008).

Geng, L., M.C. Zatko, B. Alexander, T.J. Fudge, A.J. Schauer, L.T. Murray and L.J. Mickley, Effects of post-depositional processing on nitrogen isotopes of nitrate in the Greenland Ice Sheet Project 2 (GISP 2) ice core, Geophys. Res. Lett., 42, 5346-5354, DOI: 10.1002/2015GL064218 (2015).

---

## Referee Comment (RC2) · Anonymous Referee #2 · 5 Jul 2017

I absolutely appreciate that the authors have gone to a great deal of effort to collect unique samples along a climatic gradient in Greenland. However, this manuscript needs a fair amount of work to be of publishable quality. There is a need for more digestion of the data that they have, more updated discussion of the literature surrounding interpretation of isotopes of snowpack nitrate, and a fuller interpretation of the data presented such that conclusions are drawn based upon the evidence presented.

The work presents chemical measurements of snow collected from three different sites in Western Greenland - a coastal site, an inland site, and a further inland site that represents the start of the interior ice sheet. Ionic composition of the snow samples, from

snowpack collections and snow on top of frozen lakes, are discussed in the context of differences in precipitation amount and relative proximity to oceanic (sea salt) sources. The isotopic composition of nitrate is also presented and discussed in the context of sources of atmospheric nitrate and post-depositional processing of snowpack nitrate.

The primary conclusions drawn in the study are focused on the N isotopic composition of nitrate. The authors neglect a fair amount of recent literature on the subject, do not compare and contrast with other work done in Greenland that is relevant, and do not interpret the oxygen isotopes of nitrate. The conclusion that there is a strong gradient in d15N from the margin to the coast is not supported by the evidence presented. This conclusion is based on contrasting the most interior site (-7.5 per mille) to the coastal site (-11.3 per mille). However, the midpoint site has a value higher than the interior site (-5.7 per mille) negating the use of the terminology gradient. Furthermore, there is no comparison with seasonal snowpack at Summit, Greenland (the authors appear to argue that they cannot compare with accumulated snow which I take to mean ice core). Kunasek et al. (JGR, 2008), Hastings et al. (JGR, 2004), and Fibiger et al. (JGR, 2016; GRL, 2013) all present seasonal snow results for the isotopes of nitrate. In fact, the wintertime snowpack at Summit has a mean value similar to the coastal site, i.e. -10 +/- 3.2 per mille in Hastings et al. (2004). Further, the more enriched d15N values found in late summer rain at Kellyville and the Ice Sheet also match the snowpack seasonality at Summit (summer = ∼0 per mille). This makes it highly questionable whether a gradient across the interior of Greenland to the coast can be considered at all. Instead of using a decadal or multi-annual value of -1 per mille from Hastings et al. (2009), the early spring snowpack mean values can be compared with (modern) seasonal snowpack at Summit.

The discussion on post-depositional processing needs to be made much clear in the text. First and foremost, there is no definition of "post-depositional processing". The manuscript primarily uses this term to mean "loss" of nitrate from the snow, but other times it could represent recycling of nitrate following loss from the snowpack. This

needs to be made more clear. In addition, each of the processes discussed - photolysis, volatilization, sublimation - have been discussed in the literature before and at least two of these processes have calculated fractionation factors that could be considered in the discussion (see Frey et al., ACP, 2009). Frey and others have also followed up this earlier work to study photolytic loss in the field and in the laboratory (Berhanu et al., J. Chem. Phys., 2014; Berhanu et al., ACP, 2015; Erbland et al., ACP, 2015 and references therein). The work in Antarctica may or may not be relevant in the manuscript here, but the impact of loss of nitrate from the snow on the isotopes of nitrate is made most clear by the body of work that has been completed at Dome C.

The distinction between post-depositional loss and post-depositional processing is critical. Loss should lead to enrichment of d15N of residual nitrate; pure loss should also lead to enrichment of d18O, however, several studies have shown d18O (and D17O) tend to show decreases when nitrate is lost via photolysis (see lab studies by Berhanu mentioned above, also McCabe et al., JGR, 2005). Erbland et al. (ACP, 2013) suggest that the changes in D17O are negligible; Berhanu et al's field study (ACP, 2015) could not necessarily distinguish the decrease in d18O and D17O b/c of poor precision on the isotope measurements, but the tendency is still observed. Shi et al. (ACP, 2015) do see significant decreases in d18O while d15N of snow nitrate increases with loss of nitrate in low accumulation zones of Antarctica. They explain this as most likely due to recombination effects - i.e. Photolysis of nitrate and reformation of nitrate in situ. This agrees well with the laboratory photolysis experiments of McCabe et al who show decreases in D17O due to recombination. Zatko et al. (ACP, 2016) model the loss and recycling of photolysis nitrate products in Greenland and Antarctica. They assume that any nitrate in the photic zone is photolyzed and lost as NOx in the gas phase, but this NOx can be reformed as nitrate and deposited back to the ice sheet. Fibiger et al. (JGR, 2016) give a very nice overview of the different ways in which nitrate can be impacted via loss and/or reformation of nitrate photolysis products. The reformation of nitrate and redeposition would likely change the oxygen isotopes markedly. The d15N should still be enriched given the very large fractionation associated with photolytic loss of nitrate;

but re-deposited nitrate (either in the condensed phase or gas phase) should alter the oxygen isotopic composition compared to the originally deposited atmospheric nitrate. This needs to be addressed in this study. The very high d18O and D17O in this study agree well with the spring results in Fibiger et al. (2016) so I am again struck more by how similar the results are to that found in the interior of the ice sheet than I am with some difference that reveals a supposed gradient across Greenland. Still, the authors should look more closely at the Zatko modeling results and compare and contrast with the model suggested gradient - I have some reservations about their modeling framework and whether it is at all realistic, but it does discuss the potential for a gradient in post-depositional processing/recycling of nitrate across the ice sheet.

The authors should really look more closely at all of the above studies (most of which are not currently cited) to put their data in better context. In particular, I would suggest comparing and contrasting with seasonal snowpack data from Summit, Greenland; better laying out the potential post-depositional loss vs. recycling mechanisms; interpreting the isotopic data more fully in the context of the possible post-dep processes (i.e. Both d15N and d18O, D17O); and reconsider whether a gradient actually exists between the coast and ice sheet sites and Summit. At this point, I would argue that their data suggests a lack of a gradient and surprising similarities with seasonal snowpack at Summit, which may actually negate the need for large re-distributions of nitrate suggested in the modeling study of Zatko.

One additional major comment has to do with the estimates of nitrate deposition on an annual basis. A much clearer discussion on this is needed. Can the rainfall measurements and the snowpack observation be combined to estimate deposition? The scaling up of snowpack that only represents half of the year seems unreasonable given that some of the rainfall samples show higher concentrations than ever found (on average) in the snowpack. How deposition is being defined in the manuscript is also not really clear. And why is this important to quantify (with such large error bars given the scaling)?

[Figure]

Finally, there are numerous times in the manuscript where "presumably" or "assumed" are used. This is distracting and also is representative of the fact that much in the manuscript is not evidence based discussion or drawing of conclusions based upon the observations.

Detailed comments:

Introduction, page 2, Line 31 - greater accumulation does not necessarily mean great precipitation rate/amount so this cannot be used as evidence to support a gradient in precipitation. Please clarify this.

Introduction, page 3, Line 4-5 - The Introduction and Abstract contrast in what the primary purpose of this study is/what is being tested. Please clarify.

Methods, page 3, line 23 - assume 100 m should be 100 cm?

Methods, page 4, chemical and isotopic analysis section - Please include a few more details on the isotopic method. Is the gold tube based pyrolysis of N2O used? How many repeated measures of samples do the std deviations represent (here and for the ion concentrations)? What sample sizes were run for isotopic analysis?

Section 3.4, page 8, line 5: here it is suggested that the snow was homogeneous on the lake surface. This is surprising given the earlier description of the major snow redistribution due to wind. Comparing/contrasting the snowpack and lake ice snow should be done much more carefully. I would argue that it is not at all clear whether these represent the "same" snow in any context.

Section 3.6, page 9: is it possible that the higher NH4+ values at the coast are due to the presence of birds? Several studies in the Arctic (and Antarctic) clearly indicate that bird guano can be a major source of atmospheric ammonia. This would better explain the distinct pattern for nitrate versus ammonium and sulfate. Also, it should be made clear if sulfate is in excess to ammonium. If not, than the explanation of ammonium sulfate deposition as a "cause" of higher concentration on the coast (page 12) does not

make sense.

Section 4.3, page 13: lines 10-20, need to compare with Fibiger et al. (2016) and Kunasek et al. (2008). Lines 20-29, this is highly speculative, you need more evidence. The "low" end of the D17O is not at all low compared to other measurements of atmospheric nitrate and other measurements of snowpack nitrate. Line 30-35, see comments above but there should be comparisons here with other relevant snowpack data (winter means, early spring surface snow at Summit - Hastings et al. (2004), Kunasek et al. (2008), Fibiger et al. (2016)). It is not as relevant to compare with a decadal or multi-year mean from the ice core in Hastings et al. (2009).

Page 14, lines 1-4: this does not make sense. Here it is being stated as a fact that "nitrate in ice cores reflects Northern Hemisphere pollutants," yet later it is argued that nitrate in snow in Greenland does represent sources.

Page 14, line 16: What is Fibiger and Hastings (2016)? It is not included in the reference list.

Section 4.3.1: In general this section would be much improved with a discussion of prevailing transport patterns. Would you expect different regions to contribute to the coast versus the interior sites? (For instance, transport studies for Summit and Dye 3 show distinct difference in expected source regions). And again, the discussion here is largely based upon the assumption of a regional gradient, however, it is not clear that a gradient does in fact exist. Further, there should be consideration of meteorological data during the time period of the study, rather than assuming (based on previous work) that the snow represent ∼50% of the annual precipitation.

As mentioned above, the Zatko et al. modeling study could give some context here as well. One possibility not considered here could also be that snow sourced emissions of NOx from the interior result in deposition of nitrate along the coast with a low d15N value reflecting the large photolytic fractionation.

[Figure]

Page 14, line 32: remove "while", the latter part of the sentence supports the former part.

Page 15, line 12: what is gas phase aerosol NO3-? and what is this assumption here of the difference in 15N based upon?

Section 4.3.2: the terminology throughout the manuscript needs to better reflect the difference between post-dep loss versus recycling of nitrate.

Page 16, line 20: While Geng et al. do assert this it is based upon an assumption about the NOx source d15N values. The more recent work by Walters et al. (already cited here), Fibiger and Hastings (2016) and Miller et al. (JGR, 2017) suggest very different source values than that compiled by Geng et al. making this assumption not valid.

Page 17, line 14: Morin's study was in coastal Arctic location, not on the ice sheet? Their data should be relevant for comparison to the coastal data here.

---

## Author Comment (AC2) · 1 Aug 2017

RESPONSES TO REVIEWER2

In response to comments from both reviews posted, we have added a new introduction and discussion based on additional relevant literature as suggested by the reviewers. We take the point that there is scope for confusion in our use of the term "gradient" given that the geographically central Kellyville sites represent one end of the data points in terms of deposition and isotopes. However our results show that there are very few significant differences between the two inland regions (KV, IS) while both differ from the coastal snowpack in the same direction (higher $\delta$15N and $\Delta$17O; lower

nitrate concentrations etc). We will therefore amend our title and heading accordingly to reflect instead the comparison of coastal with inland sites rather than implying a linear gradient.

RC2.1 Comparison with seasonal snowpack at Summit

We maintain that we do provide a comparison with seasonal snowpack at Summit as this forms a key part of our argument about the importance of postdepositional processing (see Discussion Section 4.1 where we cite numerous studies which measured recent firn and seasonal snowpack on the Greenland ice sheet and elsewhere, including Fibiger et al 2016). We thank Reviewer 2 for drawing our attention to the fact that our $\Delta$17O data are remarkably similar to those of Kunasek et al (2008) from snowpits at Summit, who found $\Delta$17O ranging from 22.4 ‰ in summer (compare our study where summer rainfall = 20.6-23.1 ‰ to 33.7 ‰ in winter (30.8-34.4 ‰ in our seasonal snowpack). We further note that our data are very similar to those from Alert, Canada and Barrow, Alaska reported in Morin et al (2008, 2012).

Furthermore, in Section 4.3.3 we specifically compare seasonal snowpack from Summit with our data and indeed cite the $\delta$15N value of -10 ‰ from the Hastings et al (2004) study (p17, L18). Like Reviewer 2, we were also struck more by the similarities between our seasonal snowpack and the winter/spring data from Summit snow, and further attribute differences between our seasonal snowpack (winter only) and ice core records with the fact that ice cores resolved annually include much less depleted summer precipitation; p17 lines 26-7). Hence we agree with the reviewer, and in fact do argue, that there is little evidence for spatial differences in the nitrate isotopic composition of the falling snow across the "gradient" (coast to ice sheet) but that there are spatial differences in how the snowpack nitrate is processed – hence the gradient or spatial pattern is one of differential postdepositional processing linked to snowpack accumulation and other climatic factors. However, to clarify the comparisons we will subdivide the discussion into comparisons with seasonal snow and comparisons with ice core records. The relevance of comparisons with ice core records (rather than just

modern snowpack) relates to the wider scope of our project looking at N deposition and isotopic composition as drivers of change in palaeolimnological records.

RC2.2 Postdepositional processing (with new section for Introduction)

We thank the reviewers for drawing our attention to the additional and more recent literature on postdepositional processing – particularly since we feel that it strengthens our interpretation of the data.

NEW TEXT: The processing of nitrate in deposited snowpack, termed postdepositional processing, occurs at the air-snow interface and may entail losses and in situ cycling of nitrate, with different impacts on both net deposition fluxes and isotopic fractionation depending on their relative importance (Frey et al., 2009; Geng et al., 2015; Fibiger et al., 2016). Nitrate may be released back to the atmosphere by desorption and evaporation as HNO3, often termed 'physical' losses (Mulvaney et al., 1998; Berhanu et al., 2015), or by photolysis (sometimes referred to as photodenitrification) (Frey et al., 2009). Photolysis of snowpack nitrate by UV radiation produces NOx, which may then undergo various processes which differ in relative importance depending on local conditions. NOx may be; 1. re-emitted from the snowpack and transported away from the area, depending on wind speed; 2. redeposited by dry deposition; 3. reoxidised back to nitrate and redeposited (re-adsorption or dissolution) (Frey et al., 2009). Erbland et al. (2015) define "nitrate recycling" as the net effect of nitrate photolysis (producing NOx), following atmospheric processing and oxidation to form atmospheric nitrate, and the local redeposition (wet or dry) and export of products. Recycling may also include redeposition of directly emitted HNO3 (Erbland et al., 2013). Hence both physical and photolytic processes may lead to effective net losses of nitrate from the snowpack if products are transported away from the location, but a proportion may be recycled and hence does not result in net removal from the snowpack, although such recycling can progressively modify isotopic signatures of the nitrate. Photolysis is associated with large fractionation of both N (15ÆŘ between -48 and -56 ‰ and O (18ÆŘ = -34 ‰ which both tend to increase $\delta$15N and $\delta$18O in the remaining snowpack nitrate if the

NOx produced is removed from the system (Frey et al., 2009; Erbland et al., 2013; Berhanu et al., 2015; Geng et al, 2015). In situ recycling of nitrate can also reduce $\delta$18O and $\Delta$17O due to oxygen isotope exchange with water (Frey et al., 2009; Shi et al., 2015), which has a different isotopic signature from atmospheric oxidants. This means that the negative 18$\varepsilon$ is not expressed in the residual snow nitrate and, in fact, the apparent overall oxygen isotope fractionation can be positive (between 9 and 13 ‰ Berhanu et al., 2015). However, the depth-integrated $\delta$15N remains constant if there is no net loss of nitrate, hence $\delta$15N is deemed a more reliable indicator of net postdepositional losses than oxygen isotopes (Geng et al., 2015; Zatko et al., 2016). Much smaller (only slightly negative) fractionation constants for other processes have been derived, e.g. physical release of nitrate (evaporation) but studies in the Antarctic by Erbland et al. (2013) found different experimental values at different temperatures and hence these factors are not generally transferable to regions with differing climatic regimes.

Antarctic studies have generally found photolysis to be the dominant driver of nitrate remobilisation and isotopic fractionation, while acknowledging that physical processes could play a greater role in coastal and other regions (Erbland et al., 2013; Berhanu et al., 2015). Erbland et al (2013, 2015) working in Antarctica found that fractionation in $\delta$18O and $\Delta$17O through nitrate loss and recycling was much less pronounced than $\delta$15N and either slightly positive or not significantly different from zero. Similar results for $\Delta$17O were also found experimentally by McCabe et al. (2005) and Berhanu et al. (2015). Erbland et al (2013) suggested that the small fractionation factors for $\delta$18O and $\Delta$17O in their coastal Antarctic snowpack could indicate a greater role for physical nitrate release, which does not entail oxygen exchange. Zatko et al. (2016) demonstrated that recycling of snow nitrate in Greenland, where nitrate spends a much shorter time in the photic zone, is much less than in Antarctica. They assumed that wet deposited nitrate is more likely to be embedded in the interior of snow grains whereas dry deposited nitrate on the grain surface should be more photolabile, so that in situ recycling is also a function of the form (wet vs dry) of nitrate deposited. New discussion:

role of postdepositional processing in isotopic differences between regions Since post-depositional processing occurs primarily in the photic zone of the snowpack (modelled values from 6-51cm in Greenland in the study of Zatko et al., 2016), a larger proportion of the snowpack at the inland sites must be exposed to such processing during spring, while much deeper snowpack at the coast will retain a greater proportion of unprocessed nitrate. Although dust inputs are likely to be greater at the inland sites, potentially reducing the depth of the snow photic zone, the much smaller snowpack and greater wind redistribution suggests a much greater potential overall for postdeposition processing through UV exposure and wind removal of photolysis or evaporative products than at the coast (cf. Frey et al., 2009). Frey et al (2005) found that at wind speeds of less than 3 m s-1 (as found at our coastal sites) the effects of wind-pumping were less important than diffusion; while our inland regions experience higher mean annual wind speeds of 3.6 (Kellyville) and 4.0 (ice sheet) m s-1. Furthermore, several studies of both modern snowpack and ice core nitrate (e.g. Geng et al, 2015) attribute differences in nitrate $\delta$15N to differences in snow accumulation rate, which is consistent with results of our study showing a less-transformed snowpack nitrate signal at the coast. Frey et al. (2009) also highlighted the importance of surface and wind-driven sublimation processes in the enrichment of insoluble chemical species and the removal of volatile species. Their study, like ours, indicated smaller nitrate transformations from snowpack in higher accumulating areas at the coast compared with inland, and the analysis of Zatko et al (2016) in both Antarctica and on the Greenland ice sheet found that enrichment of snowpack nitrate was greatest in areas with the lowest accumulation rates – consistent with our data from seasonal snowpack. The modelling study of Zatko et al (2016) also indicates that up to 100% of snowpack nitrate deposition in SW Greenland is primary deposition, rather than recycled. Our data, if we assume that coastal snowpack nitrate most closely represents regionally deposited precipitation nitrate, indicate an enrichment in $\delta$15N of 3.8 ‰ at the ice sheet and 5.6 ‰ at Kellyville, while $\Delta$17O is 3.0 ‰ higher at the ice sheet and 3.6 ‰ higher at Kellyville, relative to coastal snowpack. The lack of a concomitant decrease in $\delta$18O for inland snowpack

suggests the postdepositional enrichment in $\delta$15N may be due primarily to net losses from snowpack rather than in-situ recycling. Slightly higher mean values of $\delta$18O at inland locations, while not significant, are also suggestive of fractionating losses, rather than in situ recycling which would be expected to reduce $\delta$18O. Given that $\delta$15N shows an increase without a concomitant decrease in $\Delta$17O, nitrate loss rather than recycling would appear to be the dominant process at inland sites, which is consistent with the presence of a much smaller, more sublimated and wind-redistributed snowpack inland which favours removal and transport of photolytic and evaporative products rather than in situ recycling.

If the much higher $\delta$15N values inland do indeed reflect a much greater impact of postdepositional processing on the much smaller snowpack, then it follows that the initial snowpack deposition of nitrate may have been larger, but has subsequently been reduced by photolysis and evaporation, while coastal snowpack more faithfully records the initial atmospheric inputs of nitrate.

NEW TEXT - CONCLUSIONS FOR OUR STUDY

We conclude that at our inland regions, but especially at Kellyville, lower precipitation and snowpack accumulation in combination with higher wind speeds enhances both photolytic and physical (sublimation & evaporative) losses of snowpack nitrate. Since we see a significant enrichment in $\delta$15N but not in $\delta$18O (inland mean values are higher, but not significantly different from the coast) we suggest that in situ recycling is less important than net losses through photolysis and wind removal of NOx. Physical losses would also lead to $\delta$15N and $\delta$18O enrichment of remaining snowpack nitrate without affecting $\Delta$17O. However, we find significantly higher $\Delta$17O at our inland sites compared with coastal sites, in combination with higher $\delta$15N, but no difference in $\delta$18O. The higher $\Delta$17O found in the inland regions must therefore reflect a greater role for the O3 oxidation pathway as the source of snowpack nitrate, compared with the coastal sites.

RC2.3 Pollutant source regions

While we do of course acknowledge that there may be major differences in pollutant source regions across Greenland, in particular from the coast to the interior, the spatial scope of our study is very small relative to the size of the ice sheet and the modelled gradients shown by Zatko et al (2016). Hence we would argue that while differential source regions cannot be ruled out, our study areas are actually very close to each other relative to distances from source regions. Perhaps the most striking result is the similarity of our coastal isotopic data (both $\delta15N$ and $\Delta17O$ in seasonal snowpack but also in the summer rain samples) with studies much further afield including snowpack at Summit on the ice sheet, and atmospheric nitrate at Alert, Canada and Barrow, Alaska (Hastings et al., 2004; Kunasek et al., 2008; Morin et al., 2012; Fibiger et al., 2016). Hence it seems unlikely that differences in source region are a plausible explanation for the spatial differences in the isotopic signatures of deposited snow observed in our study.

The modelling study of Zatko et al (2016) also shows an increase in the proportional loss of nitrate through photolysis moving inland from the coast towards the ice sheet. While not directly applicable due to the lack of permanent snow cover in our study transect, their modelled enrichment of ice-core nitrate would be in the range 1-5 ‰ for the inland regions of our study and zero at the coast (Fig 11d in Zatko et al., 2016), which is entirely consistent with our findings in the seasonal snowpack.

RC2.4 Deposition estimates

Our new introductory text explaining the basis for the study should clarify why we feel it important to derive at least a first approximation of total deposition fluxes. We acknowledge that the lack of rainfall samples precludes an accurate assessment of deposition inputs but argue that we can suggest probable bounding values (min, max) with some caveats, given that rainfall represents about 50 % of total annual precipitation. Since the sparse rainfall chemistry data suggest that concentrations may differ from snowpack by 0.57-1.63x for nitrate, 1.42-1.72x for ammonium and 0.91-1.39% for sulphate, we will add deposition uncertainty estimates on the assumption that our rainfall data are representative of total rainfall chemistry.

Other Arctic studies of seasonal atmospheric nitrate have generally indicated lower summer concentrations than spring, when maxima are generally seen (e.g. Morin et al., 2012), although Dibb et al (2007) found nitrate maxima in June snowpack. Comparing our assumed snowpack accumulation period of October to March, the monthly data of Dibb et al (2007) show mean values over this period which are very close to the annual means presented for Summit. Hastings et al (2004) found mean nitrate concentrations were highest in spring and summer, while Burkhart et al (2004) found no clear seasonality.

Reviewer 2 Detailed comments:

RC2.5 Introduction, page 2, Line 31 - greater accumulation does not necessarily mean great precipitation rate/amount so this cannot be used as evidence to support a gradient in precipitation. Please clarify this.

Response: Text will be reworded accordingly.

RC2.6 Introduction, page 3, Line 4-5 - The Introduction and Abstract contrast in what the primary purpose of this study is/what is being tested. Please clarify.

Response: Additional introductory text has been provided and contrasting text reworded.

RC2.7 Methods, page 3, line 23 - assume 100 m should be 100 cm?

Response: No, 100 m is correct, this figure is our spacing of snowpack depth measurments around the catchment transects.

RC2.8 Methods, page 4, chemical and isotopic analysis section - Please include a few more details on the isotopic method. Is the gold tube based pyrolysis of N2O used?

How many repeated measures of samples do the std deviations represent (here and for the ion concentrations)? What sample sizes were run for isotopic analysis?

Response: $\delta$(15N) and $\delta$(18O) values were determined using the standard denitrifier method with N2O as analyte gas (Casciotti et al. 2002). $\Delta$(17O) was determined using the thermal decomposition method with O2 as analyte gas (Kaiser et al. 2007). $\delta$(15N) values have been corrected for isobaric interference of N217O (Kaiser & Röckmann, 2008), using the $\Delta$(17O) measurements. The standard deviations represent analyses in duplicate. As stated in section 2.3, we used 10 nmol of NO3– for isotopic analyses.

RC2.9 Section 3.4, page 8, line 5: here it is suggested that the snow was homogeneous on the lake surface. This is surprising given the earlier description of the major snow redistribution due to wind. Comparing/contrasting the snowpack and lake ice snow should be done much more carefully. I would argue that it is not at all clear whether these represent the "same' snow in any context.

Response: By homogenous we meant evenly distributed on the flat lake ice surface, compared with the patchy snow cover on catchment slopes. We will re-word accordingly. We agree that we may not be comparing the same snow and propose adding new text in the Discussion as follows:

It is possible that the snow accumulated on lake ice is of a different age mix than that sampled on catchment slopes, due to differential removal and redeposition during wind redistribution. Since higher concentrations of most ions were recorded in lake-ice snowpack (significant for nitrate at the coast and Kellyville) it may be hypothesized that postdepositional losses of nitrate are enhanced in snow on catchment slopes. Such a mechanism is also supported by the isotope data whereby terrestrial snowpack has generally higher $\delta$15N than lake ice snowpack, suggesting postdepositional enrichment. It is not possible to determine how snow has been redistributed in the current study (and in fact would be extremely difficult to measure in practice), but the consistent pattern for all lake catchments in all regions does suggest a common process operating

across the study region.

RC2.10 Section 3.6, page 9: is it possible that the higher NH4+ values at the coast are due to the presence of birds? Several studies in the Arctic (and Antarctic) clearly indicate that bird guano can be a major source of atmospheric ammonia. This would better explain the distinct pattern for nitrate versus ammonium and sulfate. Also, it should be made clear if sulfate is in excess to ammonium. If not, than the explanation of ammonium sulfate deposition as a "cause" of higher concentration on the coast (page 12) does not make sense.

Response: We did consider the possibility of biogenic sources of ammonium from seabird colonies, since these sites are all 2-3 km from the coast, but we have no recent data and older records indicate only very small colonies (20 pairs) within 10 km and the only sizeable colony of several hundred pairs is around 100 km away (Boertman et al., 1996). To our knowledge there are no major seabird colonies in the vicinity of the coastal sites.

We state on p12 (line 16) that NH4+ (17 mol ha–1 a–1) and non-seasalt SO42– (22 mol ha–1 a–1 as $\frac{1}{2}$ SO42–) are similar in charge equivalent terms; we have already accounted for the sea-salt excess (from sea-spray aerosols) of sulfate (total SO42– is more than 4x nssSO42–). The high correlation between NH4+ and nssSO42– in coastal snowpack suggests they are largely co-deposited. We therefore stand by our assertion that (NH4)2SO4 aerosols could contribute to the higher loads of NH4+ and SO42– at the coast.

RC2.11 Section 4.3, page 13: lines 10-20, need to compare with Fibiger et al. (2016) and Kunasek et al. (2008). Lines 20-29, this is highly speculative, you need more evidence. The "low" end of the D17O is not at all low compared to other measurements of atmospheric nitrate and other measurements of snowpack nitrate. Line 30-35, see comments above but there should be comparisons here with other relevant snowpack data (winter means, early spring surface snow at Summit - Hastings et al. (2004),

Kunasek et al. (2008), Fibiger et al. (2016)). It is not as relevant to compare with a decadal or multi-year mean from the ice core in Hastings et al. (2009).

Response: We have added additional discussion of the Fibiger and Kunasek papers – thank you for the suggestion. We stated in line 25 that the discussion of a possible role for ODEs is speculative, but other major ions certainly do indicate a much greater marine influence at the coast. Given the additional discussion of the $\Delta$17O data above we acknowledge that other factors could also affect the relative importance of different oxidants at the coast relative to inland regions. We do though feel that we should at least suggest possible mechanisms for the differences in $\Delta$17O observed. We have added additional text on comparing the seasonal snowpack from Summit, specifically for $\Delta$17O data. We have also added additional justification for comparing with ice core data.

RC2.12 Page 14, lines 1-4: this does not make sense. Here it is being stated as a fact that "nitrate in ice cores reflects Northern Hemisphere pollutants," yet later it is argued that nitrate in snow in Greenland does represent sources.

Response: We agree there is scope for confusion and will remove this sentence.

RC2.13 Page 14, line 16: What is Fibiger and Hastings (2016)? It is not included in the reference list.

Response: We have added the missing reference to the reference list (see below).

RC2.14 Section 4.3.1: In general this section would be much improved with a discussion of prevailing transport patterns. Would you expect different regions to contribute to the coast versus the interior sites? (For instance, transport studies for Summit and Dye 3 show distinct difference in expected source regions). And again, the discussion here is largely based upon the assumption of a regional gradient, however, it is not clear that a gradient does in fact exist. Further, there should be consideration of meteorological data during the time period of the study, rather than assuming (based on previous

work) that the snow represent _50% of the annual precipitation. As mentioned above, the Zatko et al. modeling study could give some context here as well. One possibility not considered here could also be that snow sourced emissions of NOx from the interior result in deposition of nitrate along the coast with a low d15N value reflecting the large photolytic fractionation.

Response: We do not have data for the prevailing transport patterns to either the coastal or inland locations, but we briefly discuss this issue in Section 4.3.1. A detailed discussion on this topic is beyond the scope of this paper, but we suggest to add the following text:

Kahl et al (1997) argue that trajectories to Summit on the ice sheet are similar to Dye 3 in south Greenland (Davidson et al., 1993), and that in winter, 94% belong to westerly transport patterns (in fact moving from SW coastal zones NE onto the ice sheet). Geng et al (2014) assume the dominance of N American pollutant sources at Summit. For our sites in SW Greenland it appears that similar long-range source areas would apply. Alternative approaches (lake sediment records of Pb isotopes) have indicated that European sources are also important contributors to pollution across the region (Bindler 2001a, b), while the modelling study of Zatko et al. (2016) suggests that our study region is an area of wind convergence with air flow mainly from the interior down to the coast. Hence there is no clear indication in the literature of the key local source regions affecting our study areas, but some evidence that coastal and inland areas are likely to be exposed to similar long-range sources. It is an interesting suggestion that snow-sourced (photolytically released) emissions from the interior with low $\delta$15N could contribute to low snowpack $\delta$15N in our study, but such a process alone would not explain the higher $\delta$15N values at Kellyville compared with those closer to the ice sheet and at the coast; we would still have to invoke postdepositional processing.

While we believe we have convincing evidence (in terms of precipitation, snow accumulation, wind speed, temperature etc) for increased postdepositional processing inland, we have no evidence to suggest there are likely to be major differences in source re-

gions for our study areas – especially given the similarities between isotopic signatures and concentrations in the coastal snowpack and from winter snowpack at Summit, and the location of our inland sites between the coast and the ice sheet. Our proposed 'gradient' of increased postdepositional processing moving inland from the coast is also entirely consistent with the modelling study of Zatko et al. (2016). See new text above under "Pollutant source regions"

RC2.15 Page 14, line 32: remove "while", the latter part of the sentence supports the former part.

Response: word removed

RC2.16 Response: Page 15, line 12: what is gas phase aerosol NO3-? and what is this assumption here of the difference in 15N based upon?

Response: As above – will add word "gas phase and aerosol", with several new supporting references added.

RC2.17 Section 4.3.2: the terminology throughout the manuscript needs to better reflect the difference between post-dep loss versus recycling of nitrate.

Response: Will be done, as per new text and revisions provided above.

RC2.18 Page 16, line 20: While Geng et al. do assert this it is based upon an assumption about the NOx source d15N values. The more recent work by Walters et al. (already cited here), Fibiger and Hastings (2016) and Miller et al. (JGR, 2017) suggest very different source values than that compiled by Geng et al. making this assumption not valid.

Response: We will remove this sentence.

RC2.19 Page 17, line 14: Morin's study was in coastal Arctic location, not on the ice sheet? Their data should be relevant for comparison to the coastal data here.

Response: Thanks for the correction. Morin et al.'s study was at Alert, Canada and

they compared with Summit data. We have amended the text accordingly. As pointed out, they are also reporting coastal data albeit from much further north. We have also added discussion of the later paper by Morin et al (2012) comparing data from Alert and Barrow (Alaska). We are again struck by the similarities with our own coastal data, indicating much larger scale regional similarities in nitrate isotopic composition and strengthening our argument about postdepositional processing as the most likely driver of spatial differences in our study.

Additional references to be added to the manuscript and cited above:

[revised manuscript text omitted]

Please also note the supplement to this comment:
https://www.biogeosciences-discuss.net/bg-2017-140/bg-2017-140-AC2-supplement.pdf

**Supplement:**

**RESPONSES TO REVIEWERS**

We thank both anonymous reviewers for their very detailed and constructive comments which have allowed us to propose major improvements to the manuscript, and which we believe strengthen our arguments for postdepositional processing as the most important mechanism driving spatial differences in snowpack isotopic composition. We have responded to both reviewers in turn and provide additional explanatory text and discussion for inclusion in a revised manuscript.

**RESPONSES TO REVIEWER 1**

**RC1.1** Curtis et al report measurements of ion concentrations and nitrate isotopes in 3 locations representing different snow accumulation regimes in western Greenland. All observations show gradients from the coast to the inland site on the ice sheet, with sea salt and sulfate concentrations highest at the coast while nitrate concentrations are highest inland. Most of their discussion focuses on nitrate and its nitrogen isotopic composition, where they conclude that postdepositional processing likely determines the observed spatial gradient. Given that the latter has been somewhat contested in the literature, such a study is important. They also provide estimates of the deposition flux of nitrate, ammonia, and sulfate at each location. The authors otherwise do not do as much analysis of the other data sets, such as the ions other than nitrate and oxygen isotopic composition of nitrate.

**Response:** In the interests of space we restricted the discussion mainly to nitrate and its isotopes, but included deposition estimates for sulfate and ammonium, given the paucity of such data in the Arctic and their relevance to linked ecological studies in the region. But see proposed new text below.

**RC1.2** Although the manuscript is well written as far as English language and grammar, it's missing some important background information making it somewhat hard to follow the analysis of the data. Some specific comments on this are below. The technical details seem scientifically sound. Abstract: The authors should start the abstract with a motivation for this study. Why should one be interested in the observed spatial gradients?

**Response**: We accept that more detailed motivation could be provided and would add this to the final manuscript. The study forms part of a larger study into the relative roles of N deposition vs. climate change in causing ecological change in Arctic lakes, as stated in lines 15-20 on page 2. The study region was selected because of the wealth of published ecological and palaeolimnological studies showing ecological change in a region which showed no evidence of climatic change for most of the 20th century. Hence we are interested in the possible role of N deposition in causing differential changes in coastal versus inland lakes, some of which are recorded in the lake sediment N isotopic record – hence our focus here on the N isotopes in snowpack. However, given the interesting spatial patterns observed here along with new discussions around postdepositional processing, we accept that further analysis and interpretation of the oxygen isotopes is merited and include further discussion proposed for the final manuscript as outlined below.

**RC1.3** Introduction: The introduction needs more background information. It is very short relative to the length of the entire paper. The introduction should present the potential sources of the observed ions in Greenland and discuss what controls the isotopic composition of nitrate. It should include a discussion of postdepositional processing, which is never really defined.

**Response**: New introductory text is provided below for the final version, including more introduction to isotopic sources, signatures and postdepositional processing.

**RC1.4** It should explicitly discuss why one should care about the observed spatial gradients, which seems to be the main motivation of the study.

**Response**: See above – related to published differences in lake sediment records between inland and coastal lakes

**RC1.5** Methods: Please state over what snow depth the snow samples were collected. Over the first 10 cm? Deeper? Shallower?

**Response**: The whole snowpack was sampled down to ground level and hence represents an integrated sample incorporating the net effects of postdepositional processing over the winter season (described as "depth-integrated" on Page 4 Line 7 in original text; see also Table 3a/b). New explanatory text will be added.

**RC1.6** Figure 1: What do the colors mean?

**Response**: Thank you for the comment. A figure legend will be added to explain the colour shading of 100 m contour intervals, ice sheet/land/sea/ and inland waters.

**RC1.7** Section 4.3.1: Provide a reference for the statement that "gas-phase aerosol $NO_3^-$ may be enriched in 15N compared to wet deposited $NO_3^-$". Also, "gas-phase aerosol $NO_3^-$" does not make sense. Nitrate is either the gas-phase or the aerosol phase (i.e., equilibrium partitioning between the two phases).

**Response**: The word "and" between "gas-phase" and "aerosol" was inadvertently omitted. Relevant references added to support this statement are Heaton (1987), Freyer (1991), Garten (1996) and Elliott et al. (2009).

**RC1.8** Section 4.3.2: This section was particularly hard to read because postdepositional processing is never defined. Many studies on ice sheets have shown that photolysis dominates postdepositional processing, but this is not even mentioned until the very end of this section. Perhaps if the authors properly introduce this process in the introduction, it will make it easier to clarify this section as well.

**Response**: We hope that we have clarified this in the new introductory text – see new section below.

**RC1.9** It would be useful to give the fractionation factors for the processes involved.

**Response**: Fractionation factors have been included in the new text below.

**RC1.10** Conclusion: Like the abstract, the conclusion focuses on the observed gradients with out explicitly stating why this matters. Again, a more thorough introduction may help with this.

Response: Again, hopefully the revised introduction will assist here and we have refocussed the conclusions to reflect the drivers of the spatial patterns observed.

**RC1.11** Some relevant references that could be included in the introduction and/or discussion and data comparison:
Kunasek, S.A., Alexander, B., E.J. Steig, M.G. Hastings, D.J. Gleason and J.C. Jarvis, Measurements and modeling of Δ17O of nitrate in snowpits from Summit, Greenland, J. Geophys. Res., 113, D24302 (2008).
Geng, L., M.C. Zatko, B. Alexander, T.J. Fudge, A.J. Schauer, L.T. Murray and L.J. Mickley, Effects of post-depositional processing on nitrogen isotopes of nitrate in the Greenland Ice Sheet Project 2 (GISP 2) ice core, Geophys. Res. Lett., 42, 5346-5354, DOI: 10.1002/2015GL064218 (2015)

Response: Thank you for the suggestions. We have consulted and added these references to the discussion, along with many others.

**RESPONSES TO REVIEWER2**

In response to comments from both reviews posted, we have added a new introduction and discussion based on additional relevant literature as suggested by the reviewers.

We take the point that there is scope for confusion in our use of the term "gradient" given that the geographically central Kellyville sites represent one end of the data points in terms of deposition and isotopes. However our results show that there are very few significant differences between the two inland regions (KV, IS) while both differ from the coastal snowpack in the same direction (higher $\delta^{15}N$ and $\Delta^{17}O$; lower nitrate concentrations etc). We will therefore amend our title and heading accordingly to reflect instead the comparison of coastal with inland sites rather than implying a linear gradient.

**RC2.1 Comparison with seasonal snowpack at Summit**

We maintain that we do provide a comparison with seasonal snowpack at Summit as this forms a key part of our argument about the importance of postdepositional processing (see Discussion Section 4.1 where we cite numerous studies which measured recent firn and seasonal snowpack on the Greenland ice sheet and elsewhere, including Fibiger et al 2016). We thank Reviewer 2 for drawing our attention to the fact that our $\Delta^{17}O$ data are remarkably similar to those of Kunasek et al (2008) from snowpits at Summit, who found $\Delta^{17}O$ ranging from 22.4 ‰ in summer (compare our study where summer rainfall = 20.6-23.1 ‰) to 33.7 ‰ in winter (30.8-34.4 ‰ in our seasonal snowpack). We further note that our data are very similar to those from Alert, Canada and Barrow, Alaska reported in Morin et al (2008, 2012).

Furthermore, in Section 4.3.3 we specifically compare seasonal snowpack from Summit with our data and indeed cite the $\delta^{15}$N value of -10 ‰ from the Hastings et al (2004) study (p17, L18). Like Reviewer 2, we were also struck more by the similarities between our seasonal snowpack and the winter/spring data from Summit snow, and further attribute differences between our seasonal snowpack (winter only) and ice core records with the fact that ice cores resolved annually include much less depleted summer precipitation; p17 lines 26-7). Hence we agree with the reviewer, and in fact do argue, that there is little evidence for spatial differences in the nitrate isotopic composition of the falling snow across the "gradient" (coast to ice sheet) but that there are spatial differences in how the snowpack nitrate is processed – hence the gradient or spatial pattern is one of differential postdepositional processing linked to snowpack accumulation and other climatic factors.

However, to clarify the comparisons we will subdivide the discussion into comparisons with seasonal snow and comparisons with ice core records. The relevance of comparisons with ice core records (rather than just modern snowpack) relates to the wider scope of our project looking at N deposition and isotopic composition as drivers of change in palaeolimnological records.

**RC2.2 Postdepositional processing (with new section for Introduction)**

We thank the reviewers for drawing our attention to the additional and more recent literature on postdepositional processing – particularly since we feel that it strengthens our interpretation of the data.

NEW TEXT:

[revised manuscript text omitted]

**RC2.3 Pollutant source regions**

While we do of course acknowledge that there may be major differences in pollutant source regions across Greenland, in particular from the coast to the interior, the spatial scope of our study is very small relative to the size of the ice sheet and the modelled gradients shown by Zatko et al (2016). Hence we would argue that while differential source regions cannot be ruled out, our study areas are actually very close to each other relative to distances from source regions. Perhaps the most striking result is the similarity of our coastal isotopic data (both $\delta^{15}N$ and $\Delta^{17}O$ in seasonal snowpack but also in the summer rain samples) with studies much further afield including snowpack at Summit on the ice sheet, and atmospheric nitrate at Alert, Canada and Barrow, Alaska (Hastings et al., 2004; Kunasek et al., 2008; Morin et al., 2012; Fibiger et al., 2016). Hence it seems unlikely that differences in source region are a plausible explanation for the spatial differences in the isotopic signatures of deposited snow observed in our study.

The modelling study of Zatko et al (2016) also shows an increase in the proportional loss of nitrate through photolysis moving inland from the coast towards the ice sheet. While not directly applicable due to the lack of permanent snow cover in our study transect, their modelled enrichment of ice-core nitrate would be in the range 1-5 ‰ for the inland regions of our study and zero at the coast (Fig 11d in Zatko et al., 2016), which is entirely consistent with our findings in the seasonal snowpack.

**RC2.4 Deposition estimates**

Our new introductory text explaining the basis for the study should clarify why we feel it important to derive at least a first approximation of total deposition fluxes. We acknowledge that the lack of rainfall samples precludes an accurate assessment of deposition inputs but argue that we can suggest probable bounding values (min, max) with some caveats, given that rainfall represents about 50 % of total annual precipitation. Since the sparse rainfall chemistry data suggest that concentrations may differ from snowpack by 0.57-1.63x for nitrate, 1.42-1.72x for ammonium and 0.91-1.39% for sulphate, we will add deposition uncertainty estimates on the assumption that our rainfall data are representative of total rainfall chemistry.

Other Arctic studies of seasonal atmospheric nitrate have generally indicated lower summer concentrations than spring, when maxima are generally seen (e.g. Morin et al., 2012), although Dibb et al (2007) found nitrate maxima in June snowpack. Comparing our assumed snowpack accumulation period of October to March, the monthly data of Dibb et al (2007) show mean values over this period which are very close to the annual means presented for Summit. Hastings et al (2004) found mean nitrate concentrations were highest in spring and summer, while Burkhart et al (2004) found no clear seasonality.

**Reviewer 2 Detailed comments:**

**RC2.5** Introduction, page 2, Line 31 - greater accumulation does not necessarily mean great precipitation rate/amount so this cannot be used as evidence to support a gradient in precipitation. Please clarify this.

**Response**: Text will be reworded accordingly.

**RC2.6** Introduction, page 3, Line 4-5 - The Introduction and Abstract contrast in what the primary purpose of this study is/what is being tested. Please clarify.

**Response**: Additional introductory text has been provided and contrasting text reworded.

**RC2.7** Methods, page 3, line 23 - assume 100 m should be 100 cm?

**Response**: No, 100 m is correct, this figure is our spacing of snowpack depth measurments around the catchment transects.

**RC2.8** Methods, page 4, chemical and isotopic analysis section - Please include a few more details on the isotopic method. Is the gold tube based pyrolysis of N2O used? How many repeated measures of samples do the std deviations represent (here and for the ion concentrations)? What sample sizes were run for isotopic analysis?

**Response**: $δ(^{15}N)$ and $δ(^{18}O)$ values were determined using the standard denitrifier method with $N_2O$ as analyte gas (Casciotti et al. 2002). $Δ(^{17}O)$ was determined using the thermal decomposition method with $O_2$ as analyte gas (Kaiser et al. 2007). $δ(^{15}N)$ values have been corrected for isobaric interference of $N_2^{17}O$ (Kaiser & Röckmann, 2008), using the $Δ(^{17}O)$ measurements. The standard deviations represent analyses in duplicate. As stated in section 2.3, we used 10 nmol of $NO_3^-$ for isotopic analyses.

**RC2.9** Section 3.4, page 8, line 5: here it is suggested that the snow was homogeneous on the lake surface. This is surprising given the earlier description of the major snow redistribution due to wind. Comparing/contrasting the snowpack and lake ice snow should be done much more carefully. I would argue that it is not at all clear whether these represent the "same' snow in any context.

**Response**: By homogenous we meant evenly distributed on the flat lake ice surface, compared with the patchy snow cover on catchment slopes. We will re-word accordingly. We agree that we may not be comparing the same snow and propose adding new text in the Discussion as follows:

It is possible that the snow accumulated on lake ice is of a different age mix than that sampled on catchment slopes, due to differential removal and redeposition during wind redistribution. Since higher concentrations of most ions were recorded in lake-ice snowpack (significant for nitrate at the coast and Kellyville) it may be hypothesized that postdepositional losses of nitrate are enhanced in snow on catchment slopes. Such a mechanism is also supported by the isotope data whereby terrestrial snowpack has generally higher $δ^{15}N$ than lake ice snowpack, suggesting postdepositional enrichment. It is not possible to determine how snow has been redistributed in the current study (and in fact would be extremely difficult to measure

in practice), but the consistent pattern for all lake catchments in all regions does suggest a common process operating across the study region.

**RC2.10** Section 3.6, page 9: is it possible that the higher NH4+ values at the coast are due to the presence of birds? Several studies in the Arctic (and Antarctic) clearly indicate that bird guano can be a major source of atmospheric ammonia. This would better explain the distinct pattern for nitrate versus ammonium and sulfate. Also, it should be made clear if sulfate is in excess to ammonium. If not, than the explanation of ammonium sulfate deposition as a "cause" of higher concentration on the coast (page 12) does not make sense.

**Response**: We did consider the possibility of biogenic sources of ammonium from seabird colonies, since these sites are all 2-3 km from the coast, but we have no recent data and older records indicate only very small colonies (20 pairs) within 10 km and the only sizeable colony of several hundred pairs is around 100 km away (Boertman et al., 1996). To our knowledge there are no major seabird colonies in the vicinity of the coastal sites.

We state on p12 (line 16) that $NH_4^+$ (17 mol $ha^{-1}$ $a^{-1}$) and non-seasalt $SO_4^{2-}$ (22 mol $ha^{-1}$ $a^{-1}$ as ½ $SO_4^{2-}$) are similar in charge equivalent terms; we have already accounted for the sea-salt excess (from sea-spray aerosols) of sulfate (total $SO_4^{2-}$ is more than 4x $nssSO_4^{2-}$). The high correlation between $NH_4^+$ and $nssSO_4^{2-}$ in coastal snowpack suggests they are largely co-deposited. We therefore stand by our assertion that $(NH_4)_2SO_4$ aerosols could contribute to the higher loads of $NH_4^+$ and $SO_4^{2-}$ at the coast.

**RC2.11** Section 4.3, page 13: lines 10-20, need to compare with Fibiger et al. (2016) and Kunasek et al. (2008). Lines 20-29, this is highly speculative, you need more evidence. The "low" end of the D17O is not at all low compared to other measurements of atmospheric nitrate and other measurements of snowpack nitrate. Line 30-35, see comments above but there should be comparisons here with other relevant snowpack data (winter means, early spring surface snow at Summit - Hastings et al. (2004), Kunasek et al. (2008), Fibiger et al. (2016)). It is not as relevant to compare with a decadal or multi-year mean from the ice core in Hastings et al. (2009).

**Response**: We have added additional discussion of the Fibiger and Kunasek papers – thank you for the suggestion. We stated in line 25 that the discussion of a possible role for ODEs is speculative, but other major ions certainly do indicate a much greater marine influence at the coast. Given the additional discussion of the $\Delta^{17}O$ data above we acknowledge that other factors could also affect the relative importance of different oxidants at the coast relative to inland regions. We do though feel that we should at least suggest possible mechanisms for the differences in $\Delta^{17}O$ observed.
We have added additional text on comparing the seasonal snowpack from Summit, specifically for $\Delta^{17}O$ data. We have also added additional justification for comparing with ice core data.

**RC2.12** Page 14, lines 1-4: this does not make sense. Here it is being stated as a fact that "nitrate in ice cores reflects Northern Hemisphere pollutants," yet later it is argued that nitrate in snow in Greenland does represent sources.

**Response**: We agree there is scope for confusion and will remove this sentence.

**RC2.13** Page 14, line 16: What is Fibiger and Hastings (2016)? It is not included in the reference list.

**Response**: We have added the missing reference to the reference list (see below).

**RC2.14** Section 4.3.1: In general this section would be much improved with a discussion of prevailing transport patterns. Would you expect different regions to contribute to the coast versus the interior sites? (For instance, transport studies for Summit and Dye 3 show distinct difference in expected source regions). And again, the discussion here is largely based upon the assumption of a regional gradient, however, it is not clear that a gradient does in fact exist. Further, there should be consideration of meteorological data during the time period of the study, rather than assuming (based on previous work) that the snow represent _50% of the annual precipitation. As mentioned above, the Zatko et al. modeling study could give some context here as well. One possibility not considered here could also be that snow sourced emissions of NOx from the interior result in deposition of nitrate along the coast with a low d15N value reflecting the large photolytic fractionation.

**Response**: We do not have data for the prevailing transport patterns to either the coastal or inland locations, but we briefly discuss this issue in Section 4.3.1. A detailed discussion on this topic is beyond the scope of this paper, but we suggest to add the following text:

Kahl et al (1997) argue that trajectories to Summit on the ice sheet are similar to Dye 3 in south Greenland (Davidson et al., 1993), and that in winter, 94% belong to westerly transport patterns (in fact moving from SW coastal zones NE onto the ice sheet). Geng et al (2014) assume the dominance of N American pollutant sources at Summit. For our sites in SW Greenland it appears that similar long-range source areas would apply. Alternative approaches (lake sediment records of Pb isotopes) have indicated that European sources are also important contributors to pollution across the region (Bindler 2001a, b), while the modelling study of Zatko et al. (2016) suggests that our study region is an area of wind convergence with air flow mainly from the interior down to the coast. Hence there is no clear indication in the literature of the key local source regions affecting our study areas, but some evidence that coastal and inland areas are likely to be exposed to similar long-range sources.

It is an interesting suggestion that snow-sourced (photolytically released) emissions from the interior with low $\delta^{15}N$ could contribute to low snowpack $\delta^{15}N$ in our study, but such a process alone would not explain the higher $\delta^{15}N$ values at Kellyville compared with those closer to the ice sheet and at the coast; we would still have to invoke postdepositional processing.

While we believe we have convincing evidence (in terms of precipitation, snow accumulation, wind speed, temperature etc) for increased postdepositional processing inland, we have no evidence to suggest there are likely to be major

differences in source regions for our study areas – especially given the similarities between isotopic signatures and concentrations in the coastal snowpack and from winter snowpack at Summit, and the location of our inland sites between the coast and the ice sheet. Our proposed 'gradient' of increased postdepositional processing moving inland from the coast is also entirely consistent with the modelling study of Zatko et al. (2016). See new text above under "Pollutant source regions"

**RC2.15** Page 14, line 32: remove "while", the latter part of the sentence supports the former part.

**Response**: word removed

**RC2.16** Response: Page 15, line 12: what is gas phase aerosol NO3-? and what is this assumption here of the difference in 15N based upon?

**Response**: As above – will add word "gas phase and aerosol", with several new supporting references added.

**RC2.17** Section 4.3.2: the terminology throughout the manuscript needs to better reflect the difference between post-dep loss versus recycling of nitrate.

**Response**: Will be done, as per new text and revisions provided above.

**RC2.18** Page 16, line 20: While Geng et al. do assert this it is based upon an assumption about the NOx source d15N values. The more recent work by Walters et al. (already cited here), Fibiger and Hastings (2016) and Miller et al. (JGR, 2017) suggest very different source values than that compiled by Geng et al. making this assumption not valid.

**Response**: We will remove this sentence.

**RC2.19** Page 17, line 14: Morin's study was in coastal Arctic location, not on the ice sheet? Their data should be relevant for comparison to the coastal data here.

**Response**: Thanks for the correction. Morin et al.'s study was at Alert, Canada and they compared with Summit data. We have amended the text accordingly. As pointed out, they are also reporting coastal data albeit from much further north. We have also added discussion of the later paper by Morin et al (2012) comparing data from Alert and Barrow (Alaska). We are again struck by the similarities with our own coastal data, indicating much larger scale regional similarities in nitrate isotopic composition and strengthening our argument about postdepositional processing as the most likely driver of spatial differences in our study.

**Additional references to be added to the manuscript and cited above:**

[revised manuscript text omitted]

Walters, W.W., Goodwin, S.R., Michalski, G. (2015a) Nitrogen Stable Isotope Composition (δ15N) of Vehicle-Emitted NOx. Env. Sci. Technol. 49, 2278-2285.

Zatko, M., Geng, L., Alexander, L., Sofen, E., Klein, K. (2016) The impact of snow nitrate photolysis on boundary layer chemistry and the recycling and redistribution of reactive nitrogen across Antarctica and Greenland in a global chemical transport model. Atmos. Chem. Phys. 16, 2819-2842.

---

## Author Response (AR1)

[revised manuscript text omitted]

**Thankyou for your additional comments, which we hope we have addressed in full. With regards to comments 2, 4 and 5 we have tried to justify our text/terminology but of course we can amend if required for the journal style.**

1. The scientific motivation of the study is not clear and specifically, the importance of spatial gradients or not. More relevant background information is needed on this, on potential sources of ions, and on postdepositional processes.

**Substantial new text has been added on source regions, source isotopic signatures and particularly on postdeposition processes to both the introduction and discussion.**

2. The analysis of the data is hard to follow therefore conclusions are not clearly supported. For instance in page 17 "…and nss-SO42- (not significant) are greater in coastal snowpack than inland, …". If differences are not significant, then the do not exists and cannot be used as argument. Same in the Conclusion section: "However the reverse is true for NH4+ and nss-SO42- with significantly higher concentrations in coastal snowpack than inland."

**Apologies but I cannot find the comment from p17 (perhaps this was from the original version which was subsequently resubmitted with a new Assoc Editor). I have tried to remove all ambiguity around where significant differences were observed, and believe that the substantial new additions to the text will clarify any prior confusion around results and discussion. With respect to the second comment from the conclusions, this sentence was included with reference to the previous sentence:** At the coast, sea salt ions dominate the accumulated snowpack, but are much less important in fresh snow. While $NO_3^-$ is the dominant ion at Summit (Dibb et al., 2007) its concentration declines from inland regions to the coast. However the reverse is true for $NH_4^+$ and nss-$SO_4^{2-}$ with significantly higher concentrations in coastal snowpack than inland.

**Am unclear on what the ambiguity is here? We are trying to show that at Summit, far from marine influence, the dominant ion in snowpack is nitrate with very low sulphate and ammonium. Moving from the centre of the ice sheet towards the coast, via the inland regions of our study, concentrations of nitrate decline while concentrations of sulphate (and nss-SO4) and ammonium increase.**

3. Some of the conclusions are based on correlations that in certain cases are weak (although significant) meaning that other processes are involved. In summary, drawing of conclusion should be based on observations and clearly explained. For instance:

a) "While chemistry and deposition show similarities to ice sheet snowpack, stable isotope data show major differences.

What similarities are you referring to, and which set of data are you using to

state that?" **Apologies, have added text to clarify here that we are comparing our snowpack chemistry and deposition estimates with other studies from Summit station on the Greenland ice sheet.**

b) Page 12. Where is the data for this?

"For non-sea salt sulfate, the pattern of snowpack concentrations is similar to total measured values, with the highest mean concentrations at the coast which are significantly greater than inland, although mean non-sea salt concentrations are all very low (1.0-1.8 µmol L-1). Non-sea salt Mg2+ is significantly lower at Kelly Ville than elsewhere, while non- sea salt Ca2+ increases significantly from the coast towards the ice sheet, presumably due to wind-blown minerogenic sources. There are no significant differences for non-sea salt K+ and negative values for non-sea salt Na+ suggest either non-sea salt sources of Cl- or snowpack losses of Na+, perhaps through preferential elution pathways. "

**Have added references to Table 4 where the non sea-salt chemistry data are presented**

**Other issues:**

4. Page 3. Paragraph 2. "total inorganic nitrogen". The word total is either redundant if you are measuring all inorganic N species or incorrect since N2 was not determined.

**The use of this term is common in the acid deposition literature, relating to both surface waters and precipitation chemistry, i.e. we are only talking about dissolved species here. However, if you feel it is necessary for this journal we can amend to "ammonium+nitrate". An alternative sometimes used is reactive N, though this can imply organic N species as well.**

5. Page 3. Remove double parentheses in (δ(15N), δ(18O) and Δ(17O))

**We agree that the older literature did not use parentheses in this way for delta notation, but the convention in more recent publications is moving towards this notation, and it has been used in other recent papers co-authored by Prof Jan Kaiser (author on this paper)**

6. Page 3. "We use the strong climatic gradient " Define

**We are referring to the previous paragraph indicating that mean precipitation at the coast is 631mm compared with 258mm inland, i.e. more than double. We have amended the text in several places to reflect this.**

7. Page 6. the common and used scientific notation in the isotope literature is "δ15N" or "δ15N-NO3-".

**Please see response to Comment 5 above**

8. Table 4. Change "stable isotopes (per mille) " to isotopic composition (‰)

**Done**

9. Page 11. "The nutrients NO3- and PO43- show an opposing pattern, with significantly lower concentrations in coastal snowpack than at inland sites and weak, negative correlations with Cl- (NO3-: r=-0.392, p<0.01; PO43-:

r=-0.277, p<0.05). "
**Done – text deleted**

10. Caption Figure 5. Complete… stable isotope composition of nitrate.
**Done**

11. There is a great number of non- standard abbreviations such as nss-SO42- ,
ODE's (What is ODEs by the way?). Please avoid those since they just
confuse readers.
**We have added the definition of ODEs (ozone depletion events) prior to the
first use of the term. We have also defined nss as "non-sea salt" which is a
commonly used abbreviation in the atmospheric chemistry and acid
deposition literature.**

**RESPONSES TO REVIEWERS**

We thank both anonymous reviewers for their very detailed and constructive comments which have allowed us to propose major improvements to the manuscript, and which we believe strengthen our arguments for postdepositional processing as the most important mechanism driving spatial differences in snowpack isotopic composition. We have responded to both reviewers in turn and provide additional explanatory text and discussion for inclusion in a revised manuscript.

**RESPONSES TO REVIEWER 1**

**RC1.1** Curtis et al report measurements of ion concentrations and nitrate isotopes in 3 locations representing different snow accumulation regimes in western Greenland. All observations show gradients from the coast to the inland site on the ice sheet, with sea salt and sulfate concentrations highest at the coast while nitrate concentrations are highest inland. Most of their discussion focuses on nitrate and its nitrogen isotopic composition, where they conclude that postdepositional processing likely determines the observed spatial gradient. Given that the latter has been somewhat contested in the literature, such a study is important. They also provide estimates of the deposition flux of nitrate, ammonia, and sulfate at each location. The authors otherwise do not do as much analysis of the other data sets, such as the ions other than nitrate and oxygen isotopic composition of nitrate.

**Response:** In the interests of space we restricted the discussion mainly to nitrate and its isotopes, but included deposition estimates for sulfate and ammonium, given the paucity of such data in the Arctic and their relevance to linked ecological studies in the region. But see proposed new text below. **DONE**

**RC1.2** Although the manuscript is well written as far as English language and grammar, it's missing some important background information making it somewhat hard to follow the analysis of the data. Some specific comments on this are below. The technical details seem scientifically sound. Abstract: The authors should start the abstract with a motivation for this study. Why should one be interested in the observed spatial gradients? **DONE**

**Response**: We accept that more detailed motivation could be provided and would add this to the final manuscript. The study forms part of a larger study into the relative roles of N deposition vs. climate change in causing ecological change in Arctic lakes, as stated in lines 15-20 on page 2. The study region was selected because of the wealth of published ecological and palaeolimnological studies showing ecological change in a region which showed no evidence of climatic change for most of the 20th century. Hence we are interested in the possible role of N deposition in causing differential changes in coastal versus inland lakes, some of which are recorded in the lake sediment N isotopic record – hence our focus here on the N isotopes in snowpack. However, given the interesting spatial patterns observed here along with new discussions around postdepositional processing, we accept that further analysis and interpretation of the oxygen isotopes is merited and include further discussion proposed for the final manuscript as outlined below.

**RC1.3** Introduction: The introduction needs more background information. It is very short relative to the length of the entire paper. The introduction should present the potential sources of the observed ions in Greenland and discuss what controls the isotopic composition of nitrate. It should include a discussion of postdepositional processing, which is never really defined.

**Response**: New introductory text is provided below for the final version, including more introduction to isotopic sources, signatures and postdepositional processing. **DONE – introduction has been substantially expanded with new section**

**RC1.4** It should explicitly discuss why one should care about the observed spatial gradients, which seems to be the main motivation of the study.

**Response**: See above – related to published differences in lake sediment records between inland and coastal lakes **DONE – new justification added**

**RC1.5** Methods: Please state over what snow depth the snow samples were collected. Over the first 10 cm? Deeper? Shallower?

**Response**: The whole snowpack was sampled down to ground level and hence represents an integrated sample incorporating the net effects of postdepositional processing over the winter season (described as "depth-integrated" on Page 4 Line 7 in original text; see also Table 3a/b). New explanatory text will be added. **DONE**

**RC1.6** Figure 1: What do the colors mean?

**Response**: Thank you for the comment. A figure legend will be added to explain the colour shading of 100 m contour intervals, ice sheet/land/sea/ and inland waters. **DONE**

**RC1.7** Section 4.3.1: Provide a reference for the statement that "gas-phase aerosol $NO_3^-$ may be enriched in 15N compared to wet deposited $NO_3^-$". Also, "gas-phase aerosol $NO_3^-$" does not make sense. Nitrate is either the gas-phase or the aerosol phase (i.e., equilibrium partitioning between the two phases).

**Response**: The word "and" between "gas-phase" and "aerosol" was inadvertently omitted. Relevant references added to support this statement are Heaton (1987), Freyer (1991), Garten (1996) and Elliott et al. (2009). **DONE**

**RC1.8** Section 4.3.2: This section was particularly hard to read because postdepositional processing is never defined. Many studies on ice sheets have shown that photolysis dominates postdepositional processing, but this is not even mentioned until the very end of this section. Perhaps if the authors properly introduce this process in the introduction, it will make it easier to clarify this section as well.

**Response**: We hope that we have clarified this in the new introductory text – see new section below. **DONE – substantial new text on postdepositional processing in both the introduction and discussion sections**

**RC1.9** It would be useful to give the fractionation factors for the processes involved.

**Response**: Fractionation factors have been included in the new text below. **DONE**

**RC1.10** Conclusion: Like the abstract, the conclusion focuses on the observed gradients with out explicitly stating why this matters. Again, a more thorough introduction may help with this.

**Response**: Again, hopefully the revised introduction will assist here and we have refocussed the conclusions to reflect the drivers of the spatial patterns observed. **DONE**

**RC1.11** Some relevant references that could be included in the introduction and/or discussion and data comparison:
Kunasek, S.A., Alexander, B., E.J. Steig, M.G. Hastings, D.J. Gleason and J.C. Jarvis, Measurements and modeling of Δ17O of nitrate in snowpits from Summit, Greenland, J. Geophys. Res., 113, D24302 (2008).
Geng, L., M.C. Zatko, B. Alexander, T.J. Fudge, A.J. Schauer, L.T. Murray and L.J. Mickley, Effects of post-depositional processing on nitrogen isotopes of nitrate in the Greenland Ice Sheet Project 2 (GISP 2) ice core, Geophys. Res. Lett., 42, 5346-5354, DOI: 10.1002/2015GL064218 (2015)

**Response**: Thank you for the suggestions. We have consulted and added these references to the discussion, along with many others. **DONE**

**RESPONSES TO REVIEWER2**
In response to comments from both reviews posted, we have added a new introduction and discussion based on additional relevant literature as suggested by the reviewers.
We take the point that there is scope for confusion in our use of the term "gradient" given that the geographically central Kellyville sites represent one end of the data points in terms of deposition and isotopes. However our results show that there are very few significant differences between the two inland regions (KV, IS) while both differ from the coastal snowpack in the same direction (higher $\delta^{15}N$ and $\Delta^{17}O$; lower nitrate concentrations etc). We will therefore amend our title and heading accordingly to reflect instead the comparison of coastal with inland sites rather than implying a linear gradient. **DONE**

**RC2.1 Comparison with seasonal snowpack at Summit**
We maintain that we do provide a comparison with seasonal snowpack at Summit as this forms a key part of our argument about the importance of postdepositional processing (see Discussion Section 4.1 where we cite numerous studies which measured recent firn and seasonal snowpack on the Greenland ice sheet and elsewhere, including Fibiger et al 2016). We thank Reviewer 2 for drawing our attention to the fact that our $\Delta^{17}O$ data are remarkably similar to those of Kunasek et al (2008) from snowpits at Summit, who found $\Delta^{17}O$ ranging from 22.4 ‰ in summer (compare our study where summer rainfall = 20.6-23.1 ‰) to 33.7 ‰ in winter (30.8-34.4 ‰ in our seasonal snowpack). We further note that our data are very similar to those from Alert, Canada and Barrow, Alaska reported in Morin et al (2008, 2012).

Furthermore, in Section 4.3.3 we specifically compare seasonal snowpack from Summit with our data and indeed cite the $\delta^{15}N$ value of -10 ‰ from the Hastings et al (2004) study (p17, L18). Like Reviewer 2, we were also struck more by the similarities between our seasonal snowpack and the winter/spring data from Summit snow, and further attribute differences between our seasonal snowpack (winter only) and ice core records with the fact that ice cores resolved annually include much less depleted summer precipitation; p17 lines 26-7). Hence we agree with the reviewer, and in fact do argue, that there is little evidence for spatial differences in the nitrate isotopic composition of the falling snow across the "gradient" (coast to ice sheet) but that there are spatial differences in how the snowpack nitrate is processed – hence the gradient or spatial pattern is one of differential postdepositional processing linked to snowpack accumulation and other climatic factors.

However, to clarify the comparisons we will subdivide the discussion into comparisons with seasonal snow and comparisons with ice core records. The relevance of comparisons with ice core records (rather than just modern snowpack) relates to the wider scope of our project looking at N deposition and isotopic composition as drivers of change in palaeolimnological records. **Have amended the section heading to reflect this**

**RC2.2 Postdepositional processing (with new section for Introduction)**
We thank the reviewers for drawing our attention to the additional and more recent literature on postdepositional processing – particularly since we feel that it strengthens our interpretation of the data.
NEW TEXT:
The processing of nitrate in deposited snowpack, termed postdepositional processing, occurs at the air-snow interface and may entail losses and in situ cycling of nitrate, with different impacts on both net deposition fluxes and isotopic fractionation depending on their relative importance (Frey et al., 2009; Geng et al., 2015; Fibiger et al., 2016). Nitrate may be released back to the atmosphere by desorption and evaporation as $HNO_3$, often termed 'physical' losses (Mulvaney et al., 1998; Berhanu et al., 2015), or by photolysis (sometimes referred to as photodenitrification) (Frey et al., 2009). Photolysis of snowpack nitrate by UV radiation produces $NO_x$, which may then undergo various processes which differ in relative importance depending on local conditions. $NO_x$ may be;
1. re-emitted from the snowpack and transported away from the area, depending on wind speed;
2. redeposited by dry deposition;
3. reoxidised back to nitrate and redeposited (re-adsorption or dissolution) (Frey et al., 2009).

Erbland et al. (2015) define "nitrate recycling" as the net effect of nitrate photolysis (producing $NO_x$), following atmospheric processing and oxidation to form atmospheric nitrate, and the local redeposition (wet or dry) and export of products. Recycling may also include redeposition of directly emitted $HNO_3$ (Erbland et al., 2013). Hence both physical and photolytic processes may lead to effective net losses of nitrate from the snowpack if products are transported away from the location, but a proportion may be recycled and hence does not result in net removal from the snowpack, although such recycling can progressively modify isotopic signatures of the nitrate.

Photolysis is associated with large fractionation of both N ($^{15}\varepsilon$ between -48 and -56 ‰) and O ($^{18}\varepsilon$ = -34 ‰) which both tend to increase $\delta^{15}N$ and $\delta^{18}O$ in the remaining snowpack nitrate if the $NO_x$ produced is removed from the system (Frey et al., 2009; Erbland et al., 2013; Berhanu et al., 2015; Geng et al, 2015). In situ recycling of nitrate can also reduce $\delta^{18}O$ and $\Delta^{17}O$ due to oxygen isotope exchange with water (Frey et al., 2009; Shi et al., 2015), which has a different isotopic signature from atmospheric oxidants. This means that the negative $^{18}\varepsilon$ is not expressed in the residual snow nitrate and, in fact, the apparent overall oxygen isotope fractionation can be positive (between 9 and 13 ‰, Berhanu et al., 2015). However, the depth-integrated $\delta^{15}N$ remains constant if there is no net loss of nitrate, hence $\delta^{15}N$ is deemed a more reliable indicator of net postdepositional losses than oxygen isotopes (Geng et al., 2015; Zatko et al., 2016). Much smaller (only slightly negative) fractionation constants for other processes have been derived, e.g. physical release of nitrate (evaporation) but studies in the Antarctic by Erbland et al. (2013) found different experimental values at different temperatures and hence these factors are not generally transferable to regions with differing climatic regimes.

Antarctic studies have generally found photolysis to be the dominant driver of nitrate remobilisation and isotopic fractionation, while acknowledging that physical processes could play a greater role in coastal and other regions (Erbland et al., 2013; Berhanu et al., 2015). Erbland et al (2013, 2015) working in Antarctica found that fractionation in $\delta^{18}O$ and $\Delta^{17}O$ through nitrate loss and recycling was much less pronounced than $\delta^{15}N$ and either slightly positive or not significantly different from zero. Similar results for $\Delta^{17}O$ were also found experimentally by McCabe et al. (2005) and Berhanu et al. (2015). Erbland et al (2013) suggested that the small fractionation factors for $\delta^{18}O$ and $\Delta^{17}O$ in their coastal Antarctic snowpack could indicate a greater role for physical nitrate release, which does not entail oxygen exchange. Zatko et al. (2016) demonstrated that recycling of snow nitrate in Greenland, where nitrate spends a much shorter time in the photic zone, is much less than in Antarctica. They assumed that wet deposited nitrate is more likely to be embedded in the interior of snow grains whereas dry deposited nitrate on the grain surface should be more photolabile, so that in situ recycling is also a function of the form (wet vs dry) of nitrate deposited. **DONE**

**New discussion: role of postdepositional processing in isotopic differences between regions**

[revised manuscript text omitted]

**RC2.3 Pollutant source regions**

While we do of course acknowledge that there may be major differences in pollutant source regions across Greenland, in particular from the coast to the interior, the spatial scope of our study is very small relative to the size of the ice sheet and the modelled gradients shown by Zatko et al (2016). Hence we would argue that while differential source regions cannot be ruled out, our study areas are actually very close to each other relative to distances from source regions. Perhaps the most striking result is the similarity of our coastal isotopic data (both $\delta^{15}N$ and $\Delta^{17}O$ in

seasonal snowpack but also in the summer rain samples) with studies much further afield including snowpack at Summit on the ice sheet, and atmospheric nitrate at Alert, Canada and Barrow, Alaska (Hastings et al., 2004; Kunasek et al., 2008; Morin et al., 2012; Fibiger et al., 2016). Hence it seems unlikely that differences in source region are a plausible explanation for the spatial differences in the isotopic signatures of deposited snow observed in our study.

The modelling study of Zatko et al (2016) also shows an increase in the proportional loss of nitrate through photolysis moving inland from the coast towards the ice sheet. While not directly applicable due to the lack of permanent snow cover in our study transect, their modelled enrichment of ice-core nitrate would be in the range 1-5 ‰ for the inland regions of our study and zero at the coast (Fig 11d in Zatko et al., 2016), which is entirely consistent with our findings in the seasonal snowpack. **DONE – new text added**

**RC2.4 Deposition estimates**

Our new introductory text explaining the basis for the study should clarify why we feel it important to derive at least a first approximation of total deposition fluxes. We acknowledge that the lack of rainfall samples precludes an accurate assessment of deposition inputs but argue that we can suggest probable bounding values (min, max) with some caveats, given that rainfall represents about 50 % of total annual precipitation. Since the sparse rainfall chemistry data suggest that concentrations may differ from snowpack by 0.57-1.63x for nitrate, 1.42-1.72x for ammonium and 0.91-1.39% for sulphate, we have added deposition uncertainty estimates (+/- 40%) on the assumption that our rainfall data are representative of total rainfall chemistry.

Other Arctic studies of seasonal atmospheric nitrate have generally indicated lower summer concentrations than spring, when maxima are generally seen (e.g. Morin et al., 2012), although Dibb et al (2007) found nitrate maxima in June snowpack. Comparing our assumed snowpack accumulation period of October to March, the monthly data of Dibb et al (2007) show mean values over this period which are very close to the annual means presented for Summit. Hastings et al (2004) found mean nitrate concentrations were highest in spring and summer, while Burkhart et al (2004) found no clear seasonality. **DONE – new text added**

**Reviewer 2 Detailed comments:**

**RC2.5** Introduction, page 2, Line 31 - greater accumulation does not necessarily mean great precipitation rate/amount so this cannot be used as evidence to support a gradient in precipitation. Please clarify this.

**Response**: Text will be reworded accordingly. . **DONE – inserted the word "also" to remove ambiguity, since the pptn gradient was already established from the AWS data**

**RC2.6** Introduction, page 3, Line 4-5 - The Introduction and Abstract contrast in what the primary purpose of this study is/what is being tested. Please clarify.

**Response**: Additional introductory text has been provided and contrasting text reworded. **DONE – new/revised text added to both abstract and introduction**

**RC2.7** Methods, page 3, line 23 - assume 100 m should be 100 cm?

**Response**: No, 100 m is correct, this figure is our spacing of snowpack depth measurements around the catchment transects. **NO CHANGE REQUIRED**

**RC2.8** Methods, page 4, chemical and isotopic analysis section - Please include a few more details on the isotopic method. Is the gold tube based pyrolysis of N2O used? How many repeated measures of samples do the std deviations represent (here and for the ion concentrations)? What sample sizes were run for isotopic analysis?

**Response**: $\delta(^{15}N)$ and $\delta(^{18}O)$ values were determined using the standard denitrifier method with $N_2O$ as analyte gas (Casciotti et al. 2002). $\Delta(^{17}O)$ was determined using the thermal decomposition method with $O_2$ as analyte gas (Kaiser et al. 2007). $\delta(^{15}N)$ values have been corrected for isobaric interference of $N_2^{17}O$ (Kaiser & Röckmann, 2008), using the $\Delta(^{17}O)$ measurements. The standard deviations represent analyses in duplicate. **DONE – above text added to methods. As stated in section 2.3, we used 10 nmol of $NO_3^-$ for isotopic analyses.**

**RC2.9** Section 3.4, page 8, line 5: here it is suggested that the snow was homogeneous on the lake surface. This is surprising given the earlier description of the major snow redistribution due to wind. Comparing/contrasting the snowpack and lake ice snow should be done much more carefully. I would argue that it is not at all clear whether these represent the "same' snow in any context.

**Response**: By homogenous we meant evenly distributed on the flat lake ice surface, compared with the patchy snow cover on catchment slopes. We will re-word accordingly. We agree that we may not be comparing the same snow and propose adding new text in the Discussion as follows:

It is possible that the snow accumulated on lake ice is of a different age mix than that sampled on catchment slopes, due to differential removal and redeposition during wind redistribution. Since higher concentrations of most ions were recorded in lake-ice snowpack (significant for nitrate at the coast and Kellyville) it may be hypothesized that postdepositional losses of nitrate are enhanced in snow on catchment slopes. Such a mechanism is also supported by the isotope data whereby terrestrial snowpack has generally higher $\delta^{15}N$ than lake ice snowpack, suggesting postdepositional enrichment. It is not possible to determine how snow has been redistributed in the current study (and in fact would be extremely difficult to measure in practice), but the consistent pattern for all lake catchments in all regions does suggest a common process operating across the study region. **DONE – above text added to discussion**

**RC2.10** Section 3.6, page 9: is it possible that the higher NH4+ values at the coast are due to the presence of birds? Several studies in the Arctic (and Antarctic) clearly indicate that bird guano can be a major source of atmospheric ammonia. This would better explain the distinct pattern for nitrate versus ammonium and sulfate. Also, it should be made clear if sulfate is in excess to ammonium. If not, than the explanation of ammonium sulfate deposition as a "cause" of higher concentration on the coast (page 12) does not make sense.

**Response**: We did consider the possibility of biogenic sources of ammonium from seabird colonies, since these sites are all 2-3 km from the coast, but we have no recent data and older records indicate only very small colonies (20 pairs) within 10 km and the only sizeable colony of several hundred pairs is around 100 km away (Boertman et al., 1996). To our knowledge there are no major seabird colonies in the vicinity of the coastal sites. **DONE – above text added to discussion**

We state on p12 (line 16) that $NH_4^+$ (17 mol ha$^{-1}$ a$^{-1}$) and non-seasalt $SO_4^{2-}$ (22 mol ha$^{-1}$ a$^{-1}$ as ½ $SO_4^{2-}$) are similar in charge equivalent terms; we have already accounted for the sea-salt excess (from sea-spray aerosols) of sulfate (total $SO_4^{2-}$ is more than 4x nss$SO_4^{2-}$). The high correlation between $NH_4^+$ and nss$SO_4^{2-}$ in coastal snowpack suggests they are largely co-deposited. We therefore stand by our assertion that $(NH_4)_2SO_4$ aerosols could contribute to the higher loads of $NH_4^+$ and $SO_4^{2-}$ at the coast. **NO CHANGE REQUIRED**

**RC2.11** Section 4.3, page 13: lines 10-20, need to compare with Fibiger et al. (2016) and Kunasek et al. (2008). Lines 20-29, this is highly speculative, you need more evidence. The "low" end of the D17O is not at all low compared to other measurements of atmospheric nitrate and other measurements of snowpack nitrate. Line 30-35, see comments above but there should be comparisons here with other relevant snowpack data (winter means, early spring surface snow at Summit - Hastings et al. (2004), Kunasek et al. (2008), Fibiger et al. (2016)). It is not as relevant to compare with a decadal or multi-year mean from the ice core in Hastings et al. (2009).

**Response**: We have added additional discussion of the Fibiger and Kunasek papers – thank you for the suggestion. We stated in line 25 that the discussion of a possible role for ODEs is speculative, but other major ions certainly do indicate a much greater marine influence at the coast. Given the additional discussion of the $\Delta^{17}O$ data above we acknowledge that other factors could also affect the relative importance of different oxidants at the coast relative to inland regions. We do though feel that we should at least suggest possible mechanisms for the differences in $\Delta^{17}O$ observed.
We have added additional text on comparing the seasonal snowpack from Summit, specifically for $\Delta^{17}O$ data. We have also added additional justification for comparing with ice core data. **DONE – see new text in sections 4.3.1, 4.3.2 and 4.3.3**

**RC2.12** Page 14, lines 1-4: this does not make sense. Here it is being stated as a fact that "nitrate in ice cores reflects Northern Hemisphere pollutants," yet later it is argued that nitrate in snow in Greenland does represent sources.

**Response**: We agree there is scope for confusion and will remove this sentence. **DONE – sentence deleted**

**RC2.13** Page 14, line 16: What is Fibiger and Hastings (2016)? It is not included in the reference list.

**Response**: We have added the missing reference to the reference list (see below). **DONE**

**RC2.14** Section 4.3.1: In general this section would be much improved with a discussion of prevailing transport patterns. Would you expect different regions to contribute to the coast versus the interior sites? (For instance, transport studies for Summit and Dye 3 show distinct difference in expected source regions). And again, the discussion here is largely based upon the assumption of a regional gradient, however, it is not clear that a gradient does in fact exist. Further, there should be consideration of meteorological data during the time period of the study, rather than assuming (based on previous work) that the snow represent _50% of the annual precipitation. As mentioned above, the Zatko et al. modeling study could give some context here as well. One possibility not considered here could also be that snow sourced emissions of NOx from the interior result in deposition of nitrate along the coast with a low d15N value reflecting the large photolytic fractionation.

**Response**: We do not have data for the prevailing transport patterns to either the coastal or inland locations, but we briefly discuss this issue in Section 4.3.1. A detailed discussion on this topic is beyond the scope of this paper, but we suggest to add the following text:

Kahl et al (1997) argue that trajectories to Summit on the ice sheet are similar to Dye 3 in south Greenland (Davidson et al., 1993), and that in winter, 94% belong to westerly transport patterns (in fact moving from SW coastal zones NE onto the ice sheet). Geng et al (2014) assume the dominance of N American pollutant sources at Summit. For our sites in SW Greenland it appears that similar long-range source areas would apply. Alternative approaches (lake sediment records of Pb isotopes) have indicated that European sources are also important contributors to pollution across the region (Bindler 2001a, b), while the modelling study of Zatko et al. (2016) suggests that our study region is an area of wind convergence with air flow mainly from the interior down to the coast. Hence there is no clear indication in the literature of the key local source regions affecting our study areas, but some evidence that coastal and inland areas are likely to be exposed to similar long-range sources. **DONE – see new text at start of section 4.3.1**

It is an interesting suggestion that snow-sourced (photolytically released) emissions from the interior with low $\delta^{15}N$ could contribute to low snowpack $\delta^{15}N$ in our study, but such a process alone would not explain the higher $\delta^{15}N$ values at Kellyville compared with those closer to the ice sheet and at the coast; we would still have to invoke postdepositional processing.

While we believe we have convincing evidence (in terms of precipitation, snow accumulation, wind speed, temperature etc) for increased postdepositional processing inland, we have no evidence to suggest there are likely to be major differences in source regions for our study areas – especially given the similarities between isotopic signatures and concentrations in the coastal snowpack and from winter snowpack at Summit, and the location of our inland sites between the coast and the ice sheet. Our proposed 'gradient' of increased postdepositional processing moving inland from the coast is also entirely consistent with the modelling study of Zatko et al. (2016). See new text above under "Pollutant source regions" **NO ADDITIONAL CHANGES, covered by new text in section 4.3.1 as above**

**RC2.15** Page 14, line 32: remove "while", the latter part of the sentence supports the former part.

**Response**: **DONE: word removed and replaced by "and"**

**RC2.16** Response: Page 15, line 12: what is gas phase aerosol NO3-? and what is this assumption here of the difference in 15N based upon?

**Response**: As above – will add word "gas phase and aerosol", with several new supporting references added. **DONE – see response to Reviewer 1**

**RC2.17** Section 4.3.2: the terminology throughout the manuscript needs to better reflect the difference between post-dep loss versus recycling of nitrate.

**Response**: Will be done, as per new text and revisions provided above.

**RC2.18** Page 16, line 20: While Geng et al. do assert this it is based upon an assumption about the NOx source d15N values. The more recent work by Walters et al. (already cited here), Fibiger and Hastings (2016) and Miller et al. (JGR, 2017) suggest very different source values than that compiled by Geng et al. making this assumption not valid.

**Response**: We will remove this sentence. **DONE**

**RC2.19** Page 17, line 14: Morin's study was in coastal Arctic location, not on the ice sheet? Their data should be relevant for comparison to the coastal data here.

**Response**: Thanks for the correction. Morin et al.'s study was at Alert, Canada and they compared with Summit data. We have amended the text accordingly. As pointed out, they are also reporting coastal data albeit from much further north. We have also added discussion of the later paper by Morin et al (2012) comparing data from Alert and Barrow (Alaska). We are again struck by the similarities with our own coastal data, indicating much larger scale regional similarities in nitrate isotopic composition and strengthening our argument about postdepositional processing as the most likely driver of spatial differences in our study. **DONE – text corrected in this paragraph but also new paragraph comparing the Alert and Barrow studies added to section 4.3.1.**

[revised manuscript text omitted]

---

## Editor Decision (ED1)

August 17, 2017

Review of bg-2017-140 Spatial variations in snowpack chemistry and isotopic composition of $NO_3^-$ along a nitrogen deposition gradient in West Greenland by Curtis et al.

Dear Dr. Curtis,

Thanks for submitting your scientific results to Biogeosciences. Even though both reviewers and I appreciate the valuable chemical and isotope data from snow packs, I also agree with them that the paper "needs a fair amount of work to be of publishable quality". **In addition** to the comments of the reviewers, there are two main concerns that I would like to emphasize:

1. The scientific motivation of the study is not clear and specifically, the importance of spatial gradients or not. More relevant background information is needed on this, on potential sources of ions, and on postdepositional processes.

2. The analysis of the data is hard to follow therefore conclusions are not clearly supported. For instance in page 17 "…and nss-SO42- (not significant) are greater in coastal snowpack than inland, …". If differences are not significant, then the do not exists and cannot be used as argument. Same in the Conclusion section: "However the reverse is true for NH4+ and nss-SO42- with significantly higher concentrations in coastal snowpack than inland."

3. Some of the conclusions are based on correlations that in certain cases are weak (although significant) meaning that other processes are involved. In summary, drawing of conclusion should be based on observations and clearly explained. For instance:
   a) "While chemistry and deposition show similarities to ice sheet snowpack, stable isotope data show major differences.

   What similarities are you referring to, and which set of data are you using to state that?"

   b) Page 12. Where is the data for this?
   "For non-sea salt sulfate, the pattern of snowpack concentrations is similar to total measured values, with the highest mean concentrations at the coast which are significantly greater than inland, although mean non-sea salt concentrations are all very low (1.0-1.8 µmol L-1). Non-sea salt Mg2+ is

significantly lower at Kelly Ville than elsewhere, while non- sea salt Ca2+ increases significantly from the coast towards the ice sheet, presumably due to wind-blown minerogenic sources. There are no significant differences for non-sea salt K+ and negative values for non-sea salt Na+ suggest either non-sea salt sources of Cl- or snowpack losses of Na+, perhaps through preferential elution pathways. "

**Other issues:**

4. Page 3. Paragraph 2. "total inorganic nitrogen". The word total is either redundant if you are measuring all inorganic N species or incorrect since N2 was not determined.

5. Page 3. Remove double parentheses in ($\delta$(15N), $\delta$(18O) and $\Delta$(17O))

6. Page 3. "We use the strong climatic gradient " Define

7. Page 6. the common and used scientific notation in the isotope literature is "$\delta^{15}N$" or "$\delta^{15}N-NO_3^-$".

8. Table 4. Change "stable isotopes (per mille) " to isotopic composition (‰)

9. Page 11. "The nutrients NO3- and PO43- show an opposing pattern, with significantly lower concentrations in coastal snowpack than at inland sites and weak, negative correlations with Cl- (NO3-: $r=-0.392$, $p<0.01$; PO43-: $r=-0.277$, $p<0.05$). "

10. Caption Figure 5. Complete… stable isotope composition of nitrate.

11. There is a great number of non- standard abbreviations such as nss-SO42- , ODE's (What is ODEs by the way?). Please avoid those since they just confuse readers.

I hope you can provide a new version based on the reviews

Yours sincerely

Silvio Pantoja
Associate Editor

---

## Author Response (AR2)

RESPONSE TO ASSOCIATE EDITOR COMMENTS

As requested by the Associate Editor, we have significantly shortened Section 1.1 of the Introduction (much of which was added in at the request of the Reviewers – but we have managed to edit down to save space). We have also made very minor edits to remove a few words elsewhere in the manuscript. All of these changes are marked below in track changes.

We hope that we have satisfied the Associate Editor and Reviewers and that the paper is now suitable for publication in Biogeosciences. If the paper is accepted for publication then we will proceed to arrange the open access data repository for the data included in the paper.

Many thanks

Chris Curtis (lead author)

[revised manuscript text omitted]